# Different aerosol effects on the daytime and nocturnal cloud-to-ground lightning in the Sichuan Basin

Haichao Wang[1,2], Yongbo Tan[1,2], Zheng Shi[1,2,3], Ning Yang[4], Tianxue Zheng[1,2]

[1]School of Atmospheric Physics, Nanjing University of Information Science & Technology, Nanjing, 210044, China

5 [2]China Meteorological Administration Aerosol-Cloud-Precipitation Key Laboratory, Nanjing University of Information Science & Technology, Nanjing, 210044, China

[3]State Key Laboratory of Severe Weather, Chinese Academy of Meteorological Sciences, Beijing 100081, China

[4]Changzhou Institute of Technology, School of Aeronautical and Mechanical Engineering/Flight, Changzhou, 213032, China

*Correspondence to*: Yongbo Tan (ybtan@ustc.edu) and Zheng Shi (002744@nuist.edu.cn)

10 **Abstract.** The effect of aerosols on lightning has been involved in many studies, but its mechanisms are complex and far from understood. The relationship between cloud-to-ground (CG) lightning and aerosols on an hourly time scale in the Sichuan Basin during 2010–2018 was investigated. The effects of aerosols, dynamics-thermodynamics factors (convective available potential energy: CAPE and vertical wind shear: SHEAR) and cloud-related factors (total column cloud liquid water: TCLW and total column cloud ice water: TCIW) on the CG lightning flashes on day and night were analysed. The diurnal variation 15 of CG lightning flashes has two peaks under clean conditions, while only one peak was found in the diurnal variation of ground flash under polluted conditions. In the early morning and night, more CG lightning flashes were found under polluted conditions, but in other periods, the difference in the CG lightning flashes between polluted and clean conditions is insignificant. Similar results were also found in the percentage of positive CG lightning flashes. At night, aerosols are positively correlated with the CG lightning flashes, and the response of CG lightning flashes to CAPE, SHEAR and TCLW is more evident under 20 high aerosol loading. In the afternoon, aerosols have no significant effects on CG lightning and its response to dynamics-thermodynamics and cloud-related factors. This difference seems to be caused by the different impacts of aerosol radiative and microphysical effects in these two periods. In the afternoon, aerosols may directly (indirectly) reduce the solar radiation reaching the ground and suppress convection through aerosol radiative effects (aerosol microphysical effects). Aerosols may also stimulate convection through microphysical effects. In this period, the two opposite effects of aerosols on convection 25 offset each other. At night, without solar radiation, the aerosol microphysical effects may play a dominant role in the entire AOD range to promote convection.

## 1 Introduction

Lightning is a kind of intense discharge phenomenon in the atmosphere that threatens the safety of human life and property and leads to significant increases in $NO_x$ in the middle and upper troposphere (Holle et al., 2019; Zhang et al., 2011; Choi et 30 al., 2005; Yair, 2018). Many observational studies have shown an important role of aerosols in affecting lightning from the



ocean to the continent (Westcott, 1995; Bell et al., 2009; Yuan et al., 2011; Altaratz et al., 2010, 2017; Thornton et al., 2017; Liu et al., 2020).

The effects of aerosols on lightning encompass two parts: microphysical and radiative effects (Altatatz et al., 2014; Fan et al., 2016; Li et al., 2017b; Kuniyal and Guleria, 2019). The microphysical effects are related to aerosols serving as cloud

condensation nuclei (CCN). Adding aerosols to clouds produces more small cloud droplets, inhibits coalescence, and delays the onset of warm-rain processes. For deep convective clouds, this process allows more liquid water to ascend into the mixed-phase region of the atmosphere through strong updrafts, forming more super-cooled water and ice particles. Freezing a large amount of liquid water releases more latent heat and invigorates convection (Rosenfeld et al., 2008; Khain and Lynn, 2009; Tao et al., 2012). The radiative effects suggest that aerosols can heat the atmospheric layer and cool the surface by absorbing

and scattering solar radiation, which increases the stability of the atmosphere and thus likely suppresses the development of convection (Kaufman et al., 2002; Koren et al., 2004, 2008; Tan et al., 2016; Li et al., 2017a). The content of super-cooled water and ice particles in clouds as well as the strength of convections are thus modulated by aerosols, and they are tightly correlated with the occurrence and progression of lightning (Takahashi, 1978; Jayaratne et al., 1983; Mansell et al., 2005; Saunders, 2008).

Generally, the relationships between aerosols loading and lightning or convective activity are complex and not fixed (Li et al., 2018; Lal et al., 2018; Dayeh et al., 2021). Some studies revealed the transition between two opposing effects of aerosols on clouds and lightning. The first is the microphysical effect which increases the convective intensity and lightning, followed by the radiative effect that becomes dominant with the increase in aerosol loading suppressing the convective intensity (Altaratz et al., 2010). The aerosol type determines the ability of aerosols in affecting cloud microphysical and solar radiation, which

further affects the relationship between aerosols and lightning (Li et al. 2017b). In central China, dominated by absorbing aerosols, a negative relationship was found between aerosols and thunderstorms as well as lightning. In contrast, in southeast China, dominated by hygroscopic aerosols, a positive relationship was found between them (Yang et al., 2013, 2014, 2016). The environmental condition can also change the direction of aerosol effects on lightning. For example, in dry northern Africa, the aerosols invigorate and suppress lightning under low and high aerosol loading conditions. However, in moist central Africa,

the aerosol invigoration effects are sustained under low and high aerosol loading conditions (Wang et al., 2018). In summary, the complex relationship between aerosols and lightning results from the superposition of aerosol microphysical and radiative effects, which are altered by the aerosol loading, type, and specific environmental conditions. Some model studies simulated the development of thunderstorms under different aerosol backgrounds and found elevating aerosol concentration leads to the enhancement of lightning even if the aerosol concentration reaches a high level (Zhao et al., 2015; Shi et al., 2015, 2019).

These model studies can be conducted to represent the detailed aerosol microphysical effects but are difficult to be revealed in observational studies due to the presence of aerosol radiative effects.

Many observational studies concerning the relationship between aerosols and lightning on a relatively coarse time



resolution (such as a yearly, monthly, or daily time scale) have been published (Lal et al., 2011; Wang et al., 2011; Altaratz et al., 2017; Shi et al., 2020; Liu et al., 2021; Wang et al., 2021; Chakraborty et al., 2021). Few studies have investigated the

linkage between aerosols and lightning (and convective activity) on an hourly time scale. Guo et al. (2016) investigated the daily cycle of precipitation and lightning under clean and polluted conditions over the Pearl River Delta. Under the polluted condition, heavy precipitation and lightning occurrences were delayed in the day. Chen et al. (2021) reported convective clouds preferential occur under polluted conditions in the morning but reverse in the afternoon through analysing data of two warm seasons in eastern China. These researches revealed a diurnal difference in aerosol effects. Still, changes in aerosol effects on

lightning or convective activity on an hourly time scale are far from understood.

The Sichuan Basin is one of the aerosol high-value centres in China (Liu et al., 2016; Ning et al., 2018). It also is one of the most lightning-active regions in China. Due to its unique topography and meteorological conditions, lightning over the Sichuan Basin occurs more frequently at night time during the warm season (Yang et al., 2015; Xia et al., 2015). Several studies have investigated the potential effects of aerosols on lightning activity in the Sichuan Basin (Zhao et al., 2020, 2022;

Shi et al., 2022). This study aims to investigate the relationship between aerosols and lightning activity on an hourly time scale and compare the difference in the possible effects of aerosols on lightning activity between day and night.

In this study, nine years (2010–2018) of measurements and reanalysis data over the Sichuan Basin were analysed to investigate the effects of aerosols, dynamics-thermodynamics factors, and cloud-related variables as well as their joint effects on cloud-to-ground (CG) lightning. This paper is organised as follows. Sect. 2 describes the datasets, region of interest, and

study methods. Results and discussion are given in Sect. 3. Summary and conclusion are presented in Sect. 4.

## 2 Data and methodology

### 2.1 Lightning data

The time series of cloud-to-ground (CG) lightning data used in this study were obtained from China Meteorological Administration (CMA) and China National Meteorological Center (CNMC). The CG lightning data were detected by the China

Lightning Detection Network (CLDN). This network is based on the ground-based Advanced Time of Arrival and Direction (ADTD) system and uses the Improved Performance through Combined Technology (IMPACT) method (Cummins et al. 1998). This system provides information about the time, latitude, longitude, polarity, and peak current of the CG flashes. CLDN comprises 357 sensors, covers almost all parts of central and eastern China, and has a detection efficiency of about 80–90% (Yang et al., 2015; Xia et al., 2015). Some data quality controls have been performed. Positive CG flashes with a peak current

of less than 15 kA are removed to eliminate the possible contaminations of cloud-to-cloud lightning (Cummins and Murphy, 2009). Additionally, only the first stroke is retained in a series of separate strokes that occurred for 1 second period within 10 km of the first stroke. The interval between two contiguous strokes is less than 0.5 seconds (Cummins et al. 1998). The CG



lightning data used in this study were calculated at a 0.5°×0.5° spatial resolution.

## 2.2 Aerosol data

The aerosol loading in this study is characterised by the aerosol optical depth (AOD). AOD (550nm) data are obtained from the Modern-Era Retrospective Analysis for Research and Application version 2 (MERRA-2). It provides dust, black carbon (BC), organic carbon (OC), and total extinction AODs at 550nm. The AOD of total aerosols is selected in this study to discuss the possible effects of aerosols on lightning activity. The spatial resolution of it is 0.5°×0.5°. MERRA-2 is the last version of global atmospheric reanalysis for the satellite era produced by NASA Global Modeling and Assimilation Office

(GMAO) using the Goddard Earth Observing System Model (GEOS) (Randles et al. 1980; Gelaro et al. 2017). This dataset assimilates AOD from Moderate Resolution Imaging Spectroradiometer (MODIS), Advanced Very High-Resolution Radiometer (AVHRR) over the ocean, and the space-based Multiangle Imaging Spectroradiometer (MISR) over bright surfaces, and ground-based Aerosol Robotic Network (AERONET) (Buchard et al. 2017).

## 2.3 Dynamics-thermodynamics and cloud-related data

In this study, we mainly select hourly reanalysis data of five factors: convective available potential energy (CAPE; units: J kg$^{-1}$), wind shear (SHEAR; units: m s$^{-1}$), total column cloud liquid water (TCLW; units: kg m$^{-2}$), total column cloud ice water (TCIW; units: kg m$^{-2}$), total column water vapour (TCWV), total cloud cover (CC), and 2m temperature (T). We choose CAPE and SHEAR to characterise the dynamics and thermodynamics conditions of the study region. The SHEAR is calculated from the hourly wind field ((U, V); units: m s$^{-1}$) at 925 and 500 hPa as follows:

$$SHEAR = \sqrt{(U_{500} - U_{925})^2 + (V_{500} - V_{925})^2} \ , \tag{1}$$

The TCLW and TCIW are selected to represent cloud-related parameters that affect the development of the lightning activity. The TCLW represents the amount of liquid water contained within clouds. Raindrops are not included in this parameter. The TCIW represents the amount of ice contained in clouds. Snow is not included in this parameter. These data are collected from the European Centre for Medium-Range Weather Forecast (ECMWF) ERA-5 reanalysis product with a spatial resolution of

0.25°×0.25°. To match the CG lightning and AOD data, these variables were interpolated onto a 0.5° spatial resolution grid. ERA-5 is the fifth generation ECMWF reanalysis for the global climate and weather. It combines model data with observations from across the world into a complete, consistent dataset through the law of physics (Hersbach et al., 2018, 2020).

## 2.4 Data selection and processing

This study investigates summer (June, July, and August) data from 2010 to 2018 over the study region. The data of CG

lightning, AOD, dynamics-thermodynamics, and cloud-related variables are all at a spatial resolution of 0.5°×0.5°. On a particular day, only grids with daily CG flashes higher than ten are retained to ensure that thunderstorms are occurring at these



grids on this day. The averaged AOD and dynamics-thermodynamics and cloud-related factors were then calculated among these grids to represent the values of these factors on this day. Finally, we got 564 samples. The samples were sorted according to their AOD values and divided into three equal sample subsets. The top third of the AOD range is labelled as polluted, and the bottom third is labelled as clean. The diurnal variations of CG flashes and other variables were then calculated in each subset. In Sect. 3.4, grids with more than ten CG lightning per day were recorded as samples to obtain enough samples. Finally, we got 11408 samples. The Pearson correlation is used in this study to characterise the linear relationship between two factors. The correlation is significant when it passes the significance test at the 0.05 level.

**2.5 Region of interest**

The black rectangle in Fig. 1a outlines the location of the study region, which is in southwest China. Fig. 1b presents its terrain of it. The spatial distributions of AOD and TCWV for 2010–2018, in the summer months, are shown in Figs. 1c and 1d. The black lines outline the specific study region. The study region is composed of low hills and plains, mostly entirely encircled by mountains (as shown in Fig. 1b). It contains the large city of Chengdu and parts of Chongqing. Intensely anthropogenic emissions, unique terrain, and the low-pressure system at 700hPa over the basin induce heavy air pollution in this region (Liu et al., 2016; Ning et al., 2018). This region also has rich water vapour, which is markedly larger than the adjacent areas.

**3 Results and Discussion**

**3.1 Diurnal variation of CG lightning**

Fig. 2 shows the diurnal variation in the mean occurrence frequency of CG lightning over the study region. CG lightning mainly occurs during nighttime (1800–600 BJT), and the number of CG flashes happens at night, accounting for about 72.9% of the total CG flashes. The peak of CG lightning occurs near midnight (2400–0100 BJT). This result is consistent with previous findings that the peak of CG lightning in the Sichuan Basin mainly happens at night during the warm season (May–September), while that in the contiguous southeastern part of China occurs in the afternoon (Xia et al. 2015). Figure 3 shows the spatial distribution of total CG lightning flashes and wind field with an interval of 3 hours. It can be found that the spatial distributions of the CG lightning flashes are markedly different between different periods of a day. In the morning, the CG lightning flashes mainly occur on the northwest side of the Sichuan Basin. As time goes by, the pattern of this spatial distribution is gradually disappearing. The CG lightning flashes are distributed throughout the basin in the afternoon. In the early night, the CG lightning flashes in the southern Sichuan Basin are gradually strengthened. After midnight, lightning concentrated on the southwest and northwest sides of the Sichuan Basin and progressively focused on the northwest side of the Sichuan Basin.

Some studies have investigated the causes of frequent convection and precipitation at night in the Sichuan Basin and provided some explanations. Many clouds in the Sichuan Basin during the day block the short-wave solar radiation from reaching the ground, which is not conducive to convection. At night, the cloud top radiates and cools, making it easy to form



convective activity (Li et al., 2008; Yu et al., 2010). Jin et al. (2012) proposed a conceptual model to explain the phenomenon of frequent convective and precipitation at night in the Sichuan Basin. The southwest flow in the lower troposphere strengthens at night, flows around the southeastern edge of the Yunnan-Guizhou Plateau, and enters the Sichuan Basin, forming a strong

cyclonic rotation conducive to ascending movement. This process also brings a large amount of water vapour into the basin. Meanwhile, the downdraft along the eastern slope of the Tibetan Plateau encountered the mass accumulation formed in the low altitude of the basin by the southwest warm and wet air transport in the late night, thus generating diabatic warming at a low level of the troposphere in the central basin. At late night, a cold advection from the Tibetan Plateau to the basin leads to a cooling in the middle troposphere over the central basin. In addition, a recent study found that the prominent diurnal inertial

oscillations of boundary layer south-southwesterly low-level jet into the Sichuan Basin may play an essential role in affecting the daily precipitation cycles in the Sichuan Basin (Zhang et al. 2019). The mechanisms behind the nocturnal convection and precipitation in the Sichuan Basin are complex and have not been comprehensively understood. However, the results of Figs. 2 and 3, as well as the findings of previous studies, indicate that convection and thunderstorms are more likely to occur in the study region at night than during the day. In the following content, we further investigate the potential influence of aerosols on

CG lightning in the study region on an hourly scale.

### 3.2 Changes in CG lightning associated with aerosols

The hourly variation in averaged total CG lightning flashes (TCG) is first examined under the clean and polluted subset (Fig. 4a). The histograms of the differences between polluted and clean days are also given (Fig. 4c). On the clean subset, the diurnal variation of TCG shows apparent two peaks. One is in the late afternoon (1600–1900 BJT), and the other appears after

midnight (2400–0300 BJT). In the polluted subset, the diurnal variation of TCG only has one apparent peak, which appears around midnight (2300–0200 BJT) and occurs an hour earlier than the clean subset. Except from 1600 to 1800 BJT, the TCG in the polluted subset are larger than that in the clean subset. The TCG difference between polluted and clean days is most remarkable at night, then in the morning, and the smallest in the afternoon (as shown in Fig. 4c). These results indicate that the changes in aerosol loading may lead to changes in the diurnal variation of CG lightning activity. The responses of the CG

lightning flashes to aerosol loading are also different at different times of the day. In the morning or night, adding aerosols tends to increase CG lightning flashes. In the afternoon, adding aerosols tends to decrease CG lightning flashes slightly. Note that the difference in the averaged CG lightning flashes between polluted and clean subsets is relatively small during the late afternoon and early night (about 1400–1900 BJT) but is relatively large during the early morning (about 0600–0900 BJT) and around midnight (about 2100–0300 BJT). This indicates the potential effects of aerosols on CG lightning may be apparent in

the early morning and around midnight but may be weak during the late afternoon and early night.

The hourly variations of the percentage of positive CG lightning flashes (PPCG) in the clean and polluted subset are also investigated (Fig. 4b). The PPCG in the clean or polluted subset means the ratio of the number of positive CG flashes to the





number of total CG flashes in the clean or polluted subset. In the clean and polluted subset, the value of PPCG is less than 10% during all periods. The PPCG is larger in the morning (about 0400–1100 BJT) than in other periods. In the clean subset, the

hourly variation of PPCG also has two apparent peaks, which occur in the morning (0900–1100 BJT) and early night (1800–2100 BJT). On polluted days, only one pronounced peak that appears in the morning (0900–1100 BJT) was found. The PPCG in the polluted subset is higher than that in the clean subset for most of the day except for early night (1700–2100 BJT) and some hours in the morning (0600–0700 BJT) and night (2300–2400 BJT). The space charge distribution of thunderstorms plays a crucial role in determining the polarities and types of lightning (Tan et al., 2014; Zhao et al., 2015). Therefore, in this

study, aerosols may not only affect the strength of convection and thunderstorms but may also affect the charge structure of thunderstorms. And this potential effect of aerosols on the charge structure of thunderstorms in the early night is contrary to that in other periods (except for a few hours: 0600–0700 BJT and 2300–2400 BJT). From the results of Fig. 3, we found that the location of CG lightning flashes over the study area differed at different times of the day. We speculate that this may be one of the causes for the inconsistent response results of PPCG to aerosol loading in different periods. Many studies have

investigated the relationship between aerosols and PPCG and reported different results. A positive relationship was reported between the PPCG and smoke from forest fires (Murray et al. 2000; Lang et al. 2006). The negative relationship between PPCG and urban aerosols (such as PM10 and $SO_2$) was also observed in many metropolitan regions, such as Southeastern Brazil (Naccarato et al. 2003), Taiwan (Kar et al. 2014), and South Korea (Kar et al. 2009). Tan et al. (2016) also found a positive relationship between PPCG and AOD in Nanjing.

200        Note that the results found in Fig. 4 are based on regional averages. In Fig. 3, we can find that the spatial distribution of CG lightning flashes in the study region is distinctly different at different times of the day. Therefore, the response of CG lightning flashes to aerosol loading may be spatially distinct in different periods. Fig. 5 presents the difference in the total CG lightning flashes in polluted and clean subsets (polluted−clean) with a spatial resolution of 0.5°×0.5°. Warm (cold) colours in the figure mean more (less) total CG lightning flashes in the polluted subset. The differences in most study regions are positive

in the early morning. The area with a relatively significant difference is concentrated in the northern part of the study region (Fig. 5a). Subsequently, before 1200 BJT, although the differences in most of the study region are still positive, their absolute values become smaller (Fig. 5b). In the afternoon (1200–1800 BJT), the differences in most study areas were negative, and their absolute values are small. In the early night (1800–2100 BJT), the differences in the southwest and northwest sides of the study area became positive and then expanded to the south of the study region (2100–2400 BJT) and expanded to the north

(2400–0600 BJT). After 2100 BJT, most study regions' differences became positive. Around midnight (2100–0300 BJT), the squares with a relatively significant difference are mainly concentrated in the south and part of the northwest of the study region (Fig. 5f, g), while in the latter half of the night (0300–0600 BJT), they are mainly concentrated in the northwest of the study region (Fig. 5h). The above results indicate that the response of CG lightning flashes to aerosol loading has apparent spatial distribution differences at different times of the day. The response of total CG lightning flashes to aerosol loading is





more significant in the early morning (0600–0900 BJT) and night (2100–0600 BJT) but changes little in other periods (1200–

        2100 BJT). This is consistent with the results found in Fig. 5.

        Based on the above results, we can find that the response of CG lightning flashes to aerosol loading in the study region is

        different at different times of the day. The most apparent difference exists between afternoon and night. Therefore, we further

        investigate the relationships between CG lightning flashes and aerosols during these two periods (period1: 1200–1800 BJT;

period2: 2100–0300 BJT), respectively (Fig. 6). In period1, the correlation (r=0.18) between them is weak. But in period2,

        AOD is positively correlated (r=0.69) with the TCG. Some previous studies reported a nonlinear relationship between aerosols

        and lightning. When the aerosol loading is below a threshold, lightning increases with the increase of aerosol loading, but

        when the aerosol loading exceeds this threshold, the increase of aerosol may no longer cause significant changes in lightning

        and even inhibit lightning. The explanation for this phenomenon is that when the aerosol loading is relatively low, the aerosol

microphysical effects may play the dominant role in promoting convection. When the aerosol loading further increases, the

        aerosol radiative effects become more marked, entangled with the aerosol microphysical effects, making the relationship

        between aerosols and lightning unclear, or the aerosol radiative effects dominate, leading to a decrease in convective intensity.

        In those studies, using AOD to character aerosol loading, this threshold is about AOD=0.3. In this paper, the aerosol loading

        of the polluted subset is more significant than 0.3, but at night, the CG lightning flashes of the polluted subset are still markedly

larger than that of the clean subset. Meanwhile, the relationship between aerosols and CG lightning flashes did not show a

        similar nonlinear relationship at night time. We speculate that this may be due to the lack of solar radiation at night, weakening

        aerosol radiative effects. The aerosol invigoration may dominate in convection at night, even under conditions with excessive

        aerosols. In the daytime, with solar radiation, the relationship between aerosols and lightning is similar to the results found in

        previous studies.

**3.3 Correlation between CG lightning and dynamics-thermodynamics and cloud-related factors**

        Besides aerosols, lightning activity is also controlled by dynamics-thermodynamics factors. Therefore, we selected two

        dynamics-thermodynamics factors in this section: CAPE and SHEAR. Meanwhile, two cloud-related factors: TCLW and

        TCIW, are also involved. Their correlations with CG lightning in two periods (period1 and period2) were investigated,

        respectively.

Fig. 7 presents the relationships between CG lightning flashes and two dynamics-thermodynamics factors. A positive

        relationship is found between CAPE and CG lightning flashes in period1 (r=0.85) and period2 (r=0.77). CAPE is a

        thermodynamics parameter that describes the potential buoyancy available to idealised rising air parcels and can denote the

        instability of the atmosphere (Riemann-Campe et al., 2009; Williams, 1992). A higher CAPE means that the atmosphere is

        more unstable and more likely to form thunderstorms. Many studies have reported a positive relationship between CAPE and

lightning activity (Dewan et al., 2018; Murugavel et al., 2014; Pawar et al., 2012). SHEAR is negatively correlated (r=−0.62)





with CG lightning flashes in period1, but a nonlinear relationship (r=0.35) between them is found in period2. The TCG first

increases with the increase of SHEAR, when the SHEAR exceeds about 6.5 m s$^{-1}$, the TCG changes little with the increase of

SHEAR. The SHEAR is the vertical shear of the horizontal wind. It affects the dynamic flow structures around and within a

deep convective cloud (Coniglio et al., 2006). Some studies have reported a negative relationship between SHEAR and

lightning (i.e., Wang et al. 2018; Zhao et al. 2020). In period1, our result is consistent with their findings. However, in period2,

we found a different result. Vertical wind shear can suppress vertical cloud development for isolated convection (Richardson

and Droegemeier, 2007). However, low-tropospheric and mid-tropospheric wind shear is critical in organising mesoscale

convection systems, especially for squall lines (Coniglio et al., 2006; Takemi, 2007). Thus, we may infer that the thunderstorm

system in the period1 is different to that in period2. In period1, thunderstorms in the study region may be dominated by isolated

convection. But in period2, lightning in the study region may mainly occur in mesoscale convective systems.

Fig. 8 shows the relationships between CG lightning flashes and two cloud-related factors. TCL is negatively correlated

(r=−0.67) with the CG lightning flashes in the period1 (Fig. 8a). Still, a nonlinear relationship is found between them in the

period2. In the period2, the CG lightning flashes increase with the increase of TCLW when the TCLW is relatively low, but

change little or even decrease with the rise of TCLW when the TCLW is relatively high. For TCIW, a nonlinear relationship

(r=0.46) between TCG and TCIW is also found in period1 (Fig. 8b). A positive relationship (r=0.94) between them is found in

period2. Generally, in period1, more TCG occurs under conditions with relatively low TCLW and high TCIW. However, in

period2, more TCG is found when TCLW and TCIW are relatively high. Lightning development mainly depends on the

noninductive electrification of the collision and separation between graupel and ice crystals in the presence of supercooled

water. The more ice particles, the stronger the lightning activity will be. But for TCLW, its effect on lightning may be more

complex. If the updraft is strong enough, more cloud liquid water can be transported above the freezing layer to produce more

supercooled water and ice particles, which promote lightning. However, if the cloud liquid water content is excessive, it may

be limited by the lack of enough updraft to transport it upward and tend to form warm rain rather than fuel lightning. This may

account for the nonlinear relationship between TCLW and TTCG in period2. In period1, the energy of convection in the study

region may come mainly from the heating of the surface by solar radiation, and clouds will alter the solar radiation reaching

the surface. The cloud liquid water content can reflect the cloud quantity. Therefore, the relationship between TCLW and

convective activity may be more complex during this period.

To further analyse the difference in the relationship between lightning and TCLW in these two periods, we further

investigate the relationship between TCLW and three factors (CAPE, CC, and T) in these two periods (Fig. 9). It can be found

that in both periods, TCLW is negatively correlated with the CAPE (Fig. 9a) and positive correlated (Fig. 9b) with the CC. In

period1, a marked decrease (r=−0.94) is found in T with the increase of TCLW. Still, this result is not found in period2 (Fig.

9c). In period2, the occurrence of convections in the study region mainly depends on the atmospheric instability caused by

topography and special meteorological conditions. Larger CAPE means a stronger updraft promotes convective activity,



ascends more cloud liquid water above the freezing layer and thus reduces the amount of cloud liquid water. This may be the

primary causal relationship between these two factors in period2, affecting lightning in the same direction. But in period1, the

occurrence of convection in the study region may depend more on the surface's heating by solar radiation. More cloud liquid

water leads to more clouds, reducing the amount of solar radiation reaching the surface, decreasing surface temperature and

increasing the atmosphere stability, decreasing the CAPE. Therefore, the negative relationship found in TCLW and CAPE in

period1 may contain two different causes, and these two processes affect the lightning in two opposite directions.

**3.4 Joint effects of aerosols and dynamics-thermodynamics and cloud-related factors on CG lightning**

It should be noted that the results shown in Fig. 7 and 8 are the general rather than precise relationship between CG

lightning flashes and a single factor over the study region. However, these results revealed marked differences in the

relationships between CG lightning flashes and these factors between period1 and period2. Generally, besides TCIW and

CAPE, the relationships between CG lightning flashes and the other two elements in period1 are different from that in period2.

Figs. 7 and 8 only discuss the response of CG lighting flashes to a single variable. However, the effects of aerosols on lightning

are generally entangled with the impact of dynamics and thermodynamics factors, and the aerosols also have substantial effects

on cloud properties. In the following contents, we further analyse the joint influence of aerosols and these factors on lightning

over the study area. CG lightning flashes are classified into 100 discrete cells by ten decile bins of a horizontal axis variable

and ten decile bins of a vertical axis variable (AOD-CAPE, AOD-SHEAR, AOD-TCLW, and AOD-TCIW), which ensures an

approximately equal sample size among the cells. The mean values of the CG lightning flashes are calculated in each cell.

Fig. 10 presents the joint dependence of the CG lightning flashes on the dynamics-thermodynamics factors (CAPE and

SHEAR) and AOD in period1 and period2. In period2 when AOD is fixed, more CG lightning flashes are found under

conditions with high CAPE (Fig. 10b) and high SHEAR (Fig. 10d). When CAPE and SHEAR are fixed, more CG lightning

flashes are found in boxes with high AOD. This suggests that in period2, aerosols, CAPE, and SHEAR affect lightning activity

in the same direction that all tend to promote lightning. But in period1, the responses of CG lightning flashes to aerosols are

insignificant (Fig. 10a and c). More CG lightning flashes are found under high CAPE and low SHEAR conditions. This

indicates that in period1, the role of SHEAR in affecting lightning activity reverses, and the dependence of the CG lightning

flashes on aerosols is reduced. In period2, aerosol loading seems to be an essential factor that alters the response of the CG

lightning flashes to CAPE and SHEAR. When AOD is relatively low, CG lightning flashes change little with the changes of

CAPE or SHEAR. But under conditions with relatively high AOD, the response of the CG lightning flashes to CAPE or

SHEAR is more marked.

Fig. 11 shows the joint dependence of the CG lightning flashes on the cloud-related factors (TCLW and TCIW) and AOD

during the two periods. The joint effects of AOD and TCLW on the CG lightning flashes are similar to that of AOD and SHEAR.

Meanwhile, in period2, the responses of CG lightning flashes to TCLW are relatively more marked under conditions with





relatively high AOD than that under conditions with relatively low AOD. In all periods, the CG lightning flash increases

monotonically as TCIW increases for all AOD but changes little as AOD increases in each TCIW bin. This indicates TCIW

positively correlates with the CG lightning flashes regardless of aerosol loading and periods. In addition, when looking into

the common roles of the TCLW or TCIW and AOD on the CG lightning, the data distribution along the diagonal (as shown by

the number in each cell). Samples with relatively high (low) TCLW or TCIW are concentrated in conditions with relatively

high (low) AOD. This indicates that TCLW and TCIW are both positively correlated with aerosols. However, in period1, this

pattern of TCIW is relatively insignificant to that in period2. By acting as CCN, adding aerosols will produce more but smaller

cloud droplets. On the one hand, smaller cloud droplets suppress rain and increase cloud lifetime and cloudiness, thus

increasing the TCLW. More small droplets are transported above the freezing layer through updraft to form more ice particles.

Besides, part of aerosols can also act as IN to directly increase the number of ice particles. These may be the potential causes

that increasing aerosol leads to an increase in TCIW. On the other hand, adding aerosols may lead to more clouds with high

albedo due to reduced droplet size. These clouds reduce the solar radiation towards the surface and increase the stability of the

atmosphere. Meanwhile, excessive aerosols may also stabilise the atmosphere through radiative effects. These two processes

decrease the convective intensity and reduce cloud ice water content. During period2, the negative effects of aerosols on the

cloud ice content may be weak due to the absence or reduction of solar radiation during this period. But in period1, the

formation and development of convection and thunderstorms may be more dependent on solar heating, and the negative effects

of the aerosols on the cloud ice water content may be relatively strong. This may be why the response of TCIW to aerosol

loading in period1 is not as marked as in period2.

Combining the results of Fig. 10 and Fig. 11, we find that changing aerosol loading will modulate the responses of CG

lightning flashes to the CAPE, SHEAR, and TCLW in period2. In general, the reactions of CG lightning flashes to CAPE,

SHEAR, and TCLW are relatively more apparent during this period when aerosol loading is relatively high. When the TCLW

is fixed, adding aerosols will reduce the cloud droplet size, and the cloud liquid water is more likely to ascend to the mixed-

phase region of the cloud to fuel lightning rather than form warm rain. On the other hand, when the CAPE and SHEAR are

fixed, adding aerosols will increase the content of cloud liquid water with smaller cloud droplet sizes to promote the

development of thunderstorms and thus produce more lightning. However, in period1, the relationship between aerosols and

lightning becomes more complicated, and the TCLW suppresses CG lightning flashes. Therefore, the aerosols seem to have

no marked effect on the response of lightning to these three factors during this period.

**4 Summary and conclusion**

In this study, the ground-based CG lightning measurements and the reanalysis data of AOD were analysed to investigate

the potential effects of aerosols on the lightning at an hourly scale over the Sichuan Basin. Two periods were further selected



with the most significant difference in the relationship between aerosols and lightning: afternoon (period1: 1200–1800 BJT)
and night (period2: 2100–0300 BJT). Then we investigated the relationship between CG lightning flashes, dynamics-
thermodynamics factors (CAPE and SHEAR) and cloud-related factors (TCLW and TCIW), and the joint effects of aerosols
and other factors on lightning during two periods.

A marked change in the diurnal variation of the CG lightning flashes was found between the clean and polluted subsets
over the study region. In the clean subset, the diurnal variation of the CG lightning flashes shows two peaks that occur in the
middle of the night and late afternoon, respectively. But in the polluted subset, the peak in the late afternoon was diminished.
Similar results were also found in PPCG. Meanwhile, more CG lightning flashes and larger PPCG are found in the polluted
subset than those in the clean subset during morning and night. In the afternoon and early night, the differences in CG lightning
flashes and PPCG between clean and polluted subsets were insignificant. In addition, the increase of CG lightning flashes in
the polluted subset was mainly concentrated in the northern parts of the study region in the morning. At night, the profound
CG lightning production is concentrated in the southern parts, the southwest side, and the northwest side of the study region.
Therefore, it can be inferred that aerosols have different effects on lightning at different times in the study region. At night and
in the early morning, aerosols may play a major role in promoting lightning in the study region, while in the afternoon, the
impact of aerosols on lightning maybe not apparent.

CAPE and TCIW promote lightning in both periods, suggesting that convective uplift and ice-phase particles are crucial
factors for lightning. SHEAR tends to suppress lightning in the afternoon and reverse at night. This indirectly indicates
differences in thunderstorm systems in the study region between the afternoon and night. In the afternoon, it may be dominated
by isolated thunderstorms, while at night, it may be overwhelmed by mesoscale convective systems. Increasing TCLW leads
to the decrease of CG lightning flashes in the afternoon but produces more CG lightning flashes at night. One can conclude
that this may be due to the different causes of convective activities in the study region between afternoon and night. On the
one hand, increasing cloud liquid water forms more supercooled water and ice particles through updraft to fuel lightning. On
the other hand, the increase of cloud liquid water will also lead to the increase of clouds, and reduce the solar radiation reaching
the surface, thus reducing the surface temperature, increasing the stability of the atmosphere, and inhibiting convection. In the
afternoon, the convective activity in the study area may depend on solar radiation, and the negative effect of cloud liquid water
on convective activity seems to be dominant. At night, the influence of cloud liquid water on solar radiation is absent, and
cloud liquid water may play a major role in promoting lightning.

In addition, we found that at night, lightning response to CAPE, SHEAR and TCLW is more evident under the condition
of high aerosol loading. In contrast, in the afternoon, aerosols have no noticeable effect on the relationship between these
factors and lightning. Figure 12 shows a schematic diagram explaining the potential role of aerosols in these two periods. Due
to the unique topography and meteorological conditions, the convection and thunderstorms in Sichuan Basin are more active
at night (especially at midnight) and morning (especially in the early morning). In the afternoon, the atmosphere is relatively

stable and not prone to the formation of convection and thunderstorms. In this period, the positive and negative effects of aerosols on convection and thunderstorms may both exist. On the one hand, adding aerosols may lead to more cloud liquid water due to reduced droplet size and a narrower droplet spectrum. Increasing cloud liquid water may lead to two opposite effects on lightning as described above. On the other hand, under a clear sky or a sky with thin clouds, excessive aerosols may

stabilise the atmosphere through radiative effects. These effects of aerosols on convection and thunderstorms may entangle each other and complicate the relationship between aerosols and lightning. Therefore, in this period, the overall effects of aerosols on lightning are insignificant. At night or early morning, the negative impact of aerosols on convection and thunderstorms may be weak due to the absence or decrease of solar radiation. Aerosols may play the dominant role in invigorating convection and thunderstorms through microphysical effects. In this period, increasing aerosols leads to more but

small cloud droplets with fixed cloud liquid content, thus inhibiting the warm-rain process. So, more cloud liquid water can ascend to the freezing layer of the atmosphere to invigorate lightning under the same dynamics-thermodynamics condition. Therefore, in this period, lightning is positively correlated with aerosol and the effects of CAPE, SHEAR, and TCLW are more apparent under conditions with more aerosols.

Moreover, the analysis of this study is based on limited measurements and reanalysis data. The mechanisms behind these

phenomena still need further research through analysis based on more comprehensive data and model simulation.

**Data availability**

Processed data in the study and the CG lightning data are available from the first author upon request (whcis4@163.com). MERRA-2 aerosol data can be downloaded from https://disc.gsfc.nasa.gov/datasets?page=1&keywords=MERRA (last access: 5 August 2022) and ERA5 data can be downloaded from https://cds.climate.copernicus.eu/#!/search?text=ERA5 (last access:

5 August 2022.)

**Author contributions**

HW, YT designed the research ideas for this study. HW carried the study out and prepared the paper. ZS and NY provided the analysis ideas for the dynamics-thermodynamics and cloud-related parameters. TZ edited the paper.

**Competing interests**

The authors declare that they have no conflict of interest.

**Acknowledgements**

This research was supported by the National Key Research and Development Program of China (2017YFC1501504),



the National Natural Science Foundation of China (Grant No. 41875003), the Open Research Program of the State Key Laboratory of Severe Weather (2019LASW-A03), and the Open Grants of the State Key Laboratory of Severe Weather (grant number 2021LASW-B05).

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





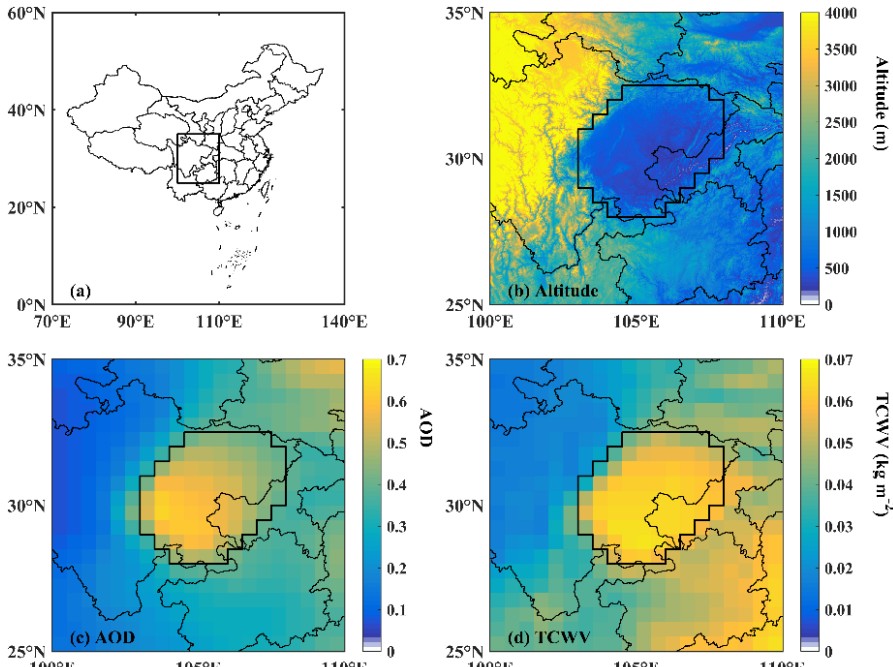

**Figure 1: (a) The location of the study region (outlined with a black rectangle). (b) The terrain of the study region is on a 0.02°×0.02° grid. Spatial distributions of (c) aerosol optical depth (AOD) and (b) total column water vapour (TCWV) from ECMWF at a spatial resolution of 0.5°×0.5° for the period 2010–2018 including the summer months (June, July, and August). The black lines in (b–d) outline the specific area investigated in this study.**

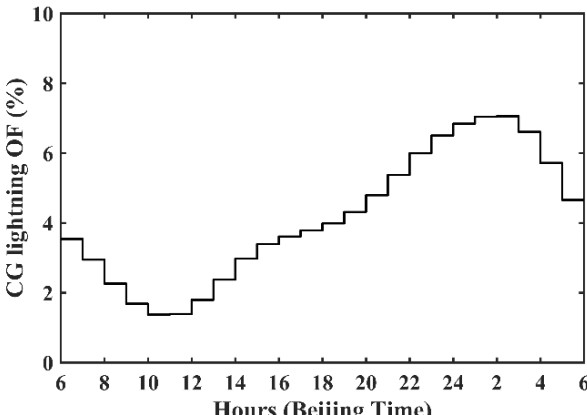

**Figure 2: The diurnal variation in the occurrence frequency of total cloud-to-ground (CG) lightning (unit: %) over the study region during the summer months (June, July, and August) of 2010–2018.**





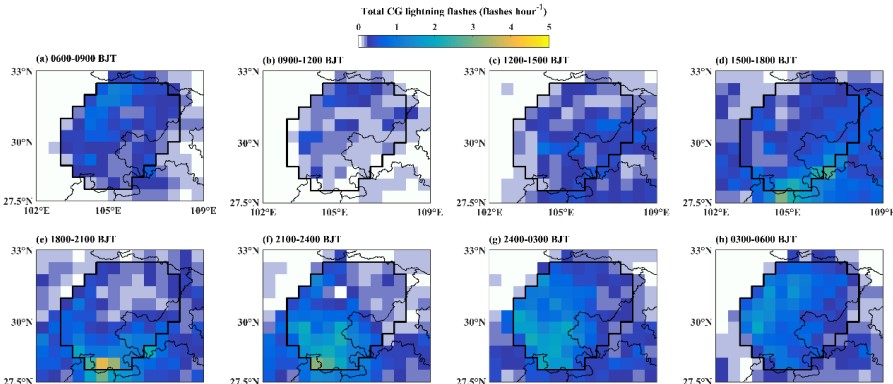

**Figure 3: Diurnal cycle of total CG lightning flashes (unit: flashes hour⁻¹) on a 0.5°×0.5° grid for 2010–2018 including the summer months (June, July, and August). The black lines outline the study region.**

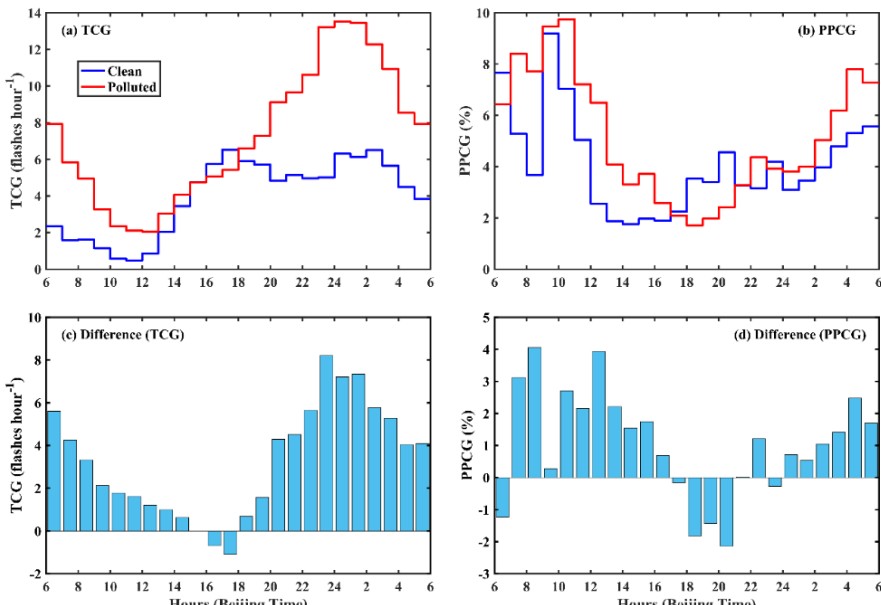

**Figure 4: The diurnal variations in (a) total CG lightning flashes (TCG), (b) negative CG lightning flashes (NCG), (c) positive CG lightning flashes (PCG), and (d) the percentage of positive CG lightning flashes (PPCG) under clean (solid blue lines) and polluted (solid red lines) days over the study region during the summer seasons of 2010-2018. The histogram in each panel represents the difference between them on polluted and clean days.**





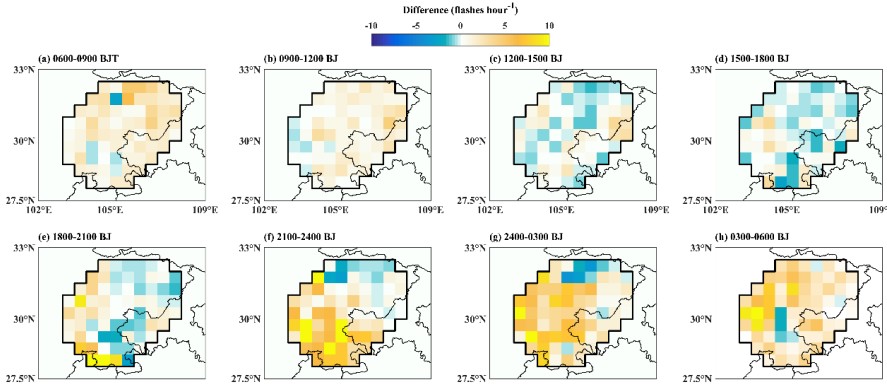

**Figure 5: Diurnal changes of total CG lightning flash differences (unite: flashes hour⁻¹) between polluted and clean subset (polluted−clean) during the study period with an interval of 3 hours (BJT). Black lines outline the study region. The spatial resolution is 0.1°×0.1°.**

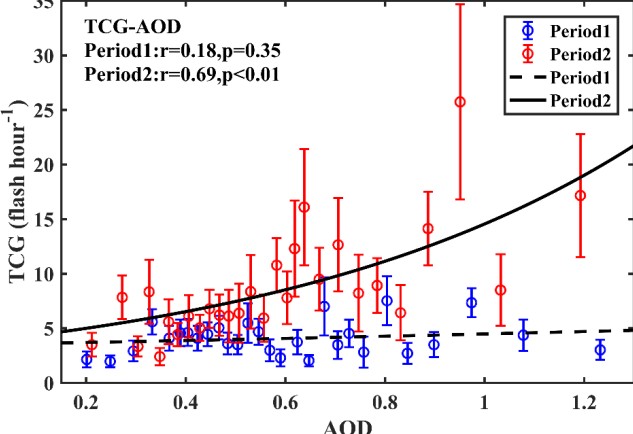

**Figure 6: Relationships between total CG lightning flashes (TCG) and AOD in (blue) period1 (1200–1800 BJT) and (red) period2 (2100–0300 BJT). Note that samples are first sorted by AOD and then every 20 points with similar AOD were averaged to create the presented scatter plot. An estimation of the error was calculated from the standard deviation of each bin divided by the square root of the number of data points in the bin. Exponential-fit curves, Pearson correlation coefficients (r), and significant level (p) are also shown.**

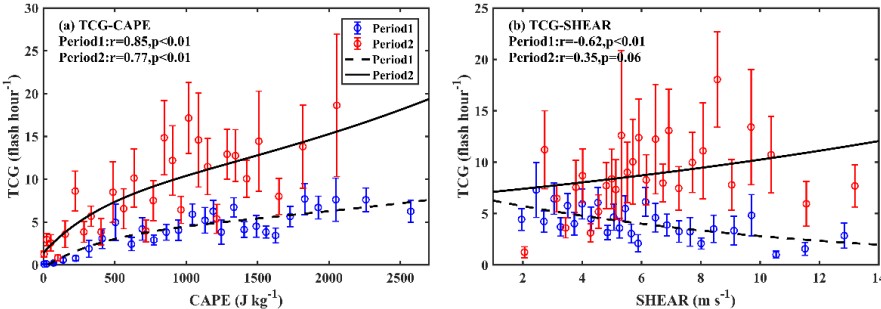

**Figure 7: Relationships between total CG lightning flashes (TCG) and two dynamics-thermodynamics factors: (a) convective available potential energy (CAPE) and (b) vertical wind shear (SHEAR) in (blue) period1 (1200–1800 BJT) and (red) period2 (2100–**





610 **0300 BJT). Note that samples are first sorted by CAPE or SHEAR and then every 20 points with similar CAPE or SHEAR were**
**averaged to create the presented scatter plot. An estimation of the error was calculated from the standard deviation of each bin**
**divided by the square root of the number of data points in the bin. Exponential-fit curves, Pearson correlation coefficients (r), and**
**significant level (p) are also shown in each panel.**

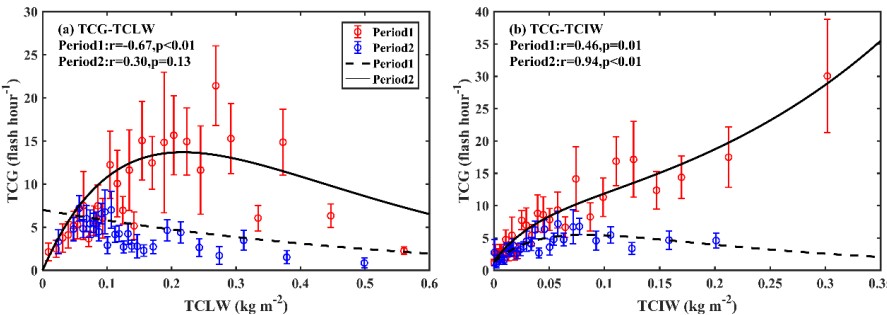

615 **Figure 8: Same as in Fig. 7, but for the (a) total column cloud liquid water (TCLW) and (b) total column ice water (TCIW).**

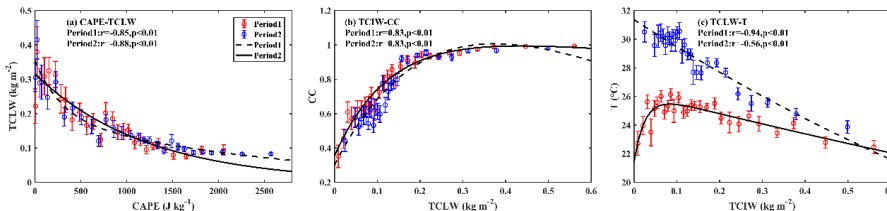

**Figure 9: Relationships in (a) total column liquid water (TCLW)-convective available potential energy (CAPE), (b) TCLW-total**
**column cover (CC), and (c) TCLW-2m temperature (T) in (blue) period1 (1200–1800 BJT) and (red) period2 (2100–0300 BJT). Note**
**that samples are first sorted by (a) CAPE or (b and c) TCLW and then every 20 points with similar CAPE or SHEAR were averaged**
620 **to create the presented scatter plot. An estimation of the error was calculated from the standard deviation of each bin divided by**
**the square root of the number of data points in the bin. Exponential-fit curves, Pearson correlation coefficients (r), and significant**
**level (p) are also shown in each panel.**

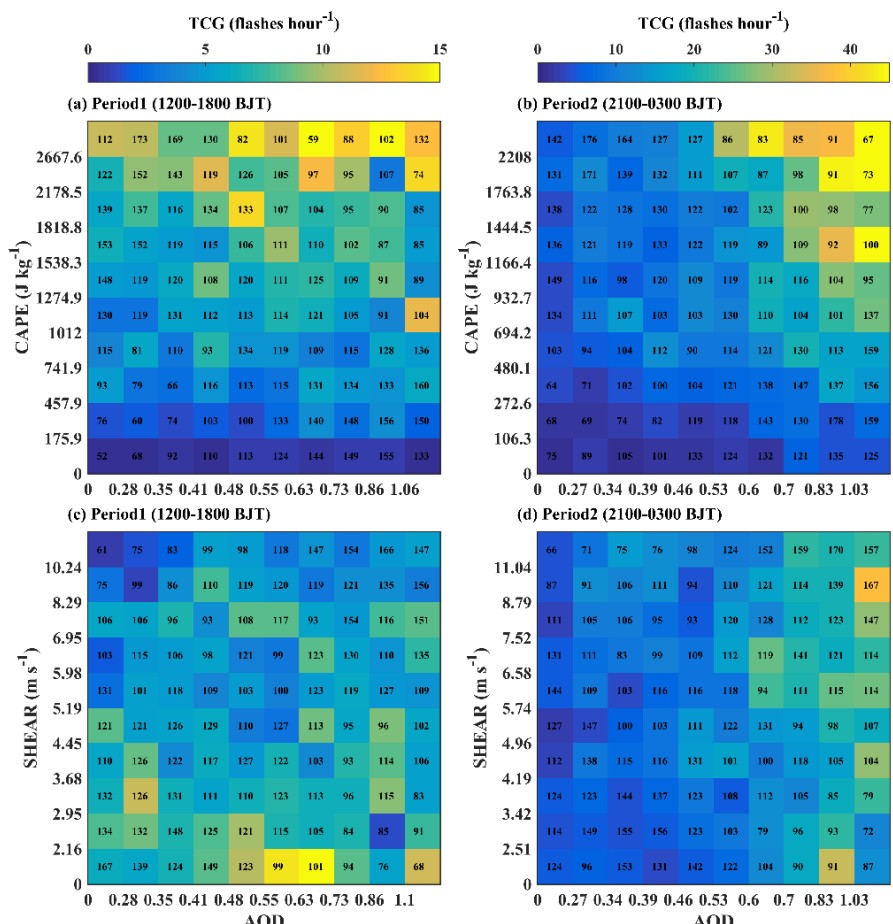

Figure 10: Joint dependence of the total CG lightning flashes (TCG) on AOD and dynamics-thermodynamics factors (CAPE and SHEAR) in (a, c) period1 (1200–1800 BJT) and (b, d) period2 (2100–0300 BJT). The number in each cell indicates the number of samples in the cell. The colour bars denote the number of total CG lightning flashes averaged in each cell.





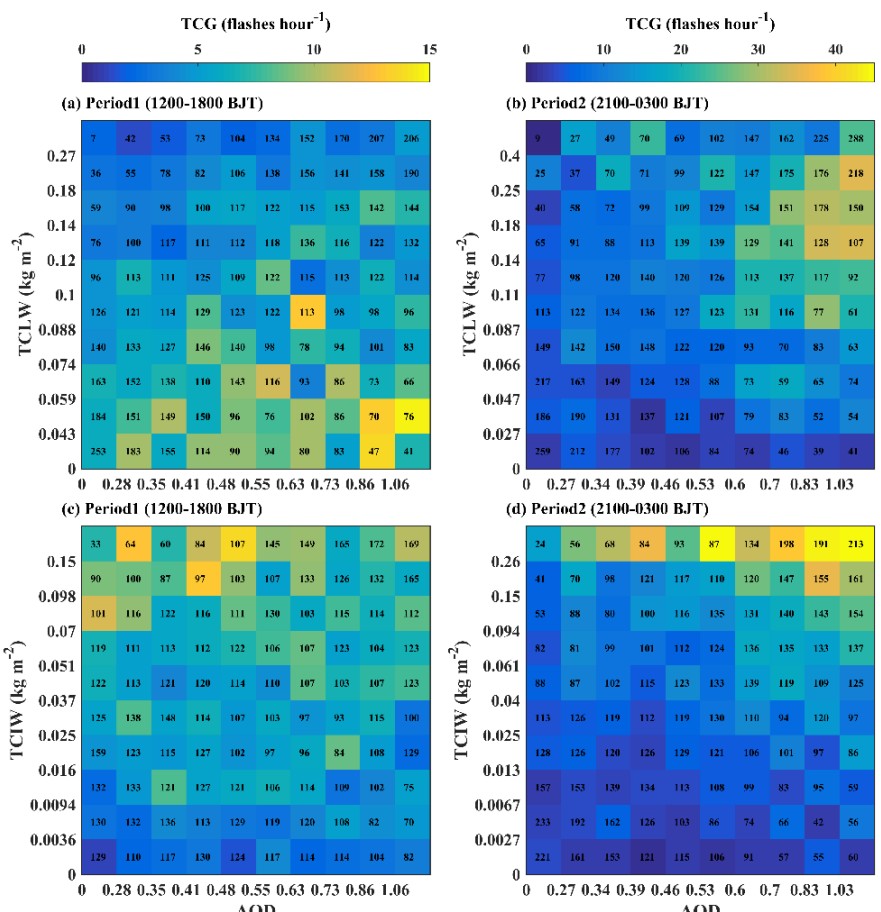

**Figure 11: Joint dependence of the total CG lightning flashes (TCG) on AOD and cloud-related factors (TCLW and TCIW) in (a, c) period1 (1200–1800 BJT) and (b, d) period2 (2100–0300 BJT). The number in each cell indicates the number of samples in the cell. The colour bars denote the number of total CG lightning flashes averaged in each cell.**

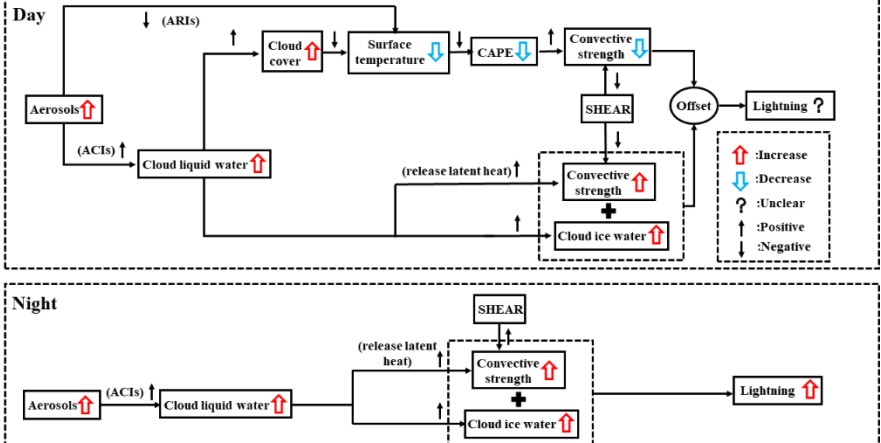

**Figure 12: Schematic diagram illustrating the effects of aerosols on lightning activity over the study region.**