# Peer review of "Diurnal differences in the effect of aerosols on cloud-to-ground lightning in the Sichuan Basin"

_Atmospheric Chemistry and Physics, 2022_

## Referee Comment (RC2)

The authors analyzed the effects aerosol on lightning during daytime and nighttime in the Sichuan Basin and tried to explain the reasons for the differences in the effects. This manuscript has some scientific significance, but there are some major comments that need to be addressed by the authors.

- There is no outstanding innovation in this paper, including analysis methods and conclusions. The authors are requested to extract the innovative points of this paper in the introduction and emphasize the innovative points in the conclusion, on the premise of adjusting the research content.
- 2. Line 81. In this paper, the data with 0.5° spatial resolution are selected to discuss the relationship between aerosol and lightning activity in the Sichuan Basin. Is it statistically significant to analyze the data with such rough resolution in such a limited space?
- 3. It is suggested to use the satellite lightning data to verify the ground-based lightning data used in this paper.
- 4. Line 95. First, there is an obvious error. The spatial resolution of the AOD data of MERRA-2 is not 0.5°×0.5°, please check and confirm. Since the resolution is not 0.5°, how to match and discuss with other data is a major problem. Secondly, the AOD data selected in this paper are reanalysis data. In the study area, the authors did not compare with the satellite observation and ground-based observation, so it is obviously unreasonable not to verify the availability of the data.
- 5. Line 105. What is the time resolution of the thermodynamic and cloud-related data selected in this paper? Please clarify. TCIW and TCLW are reanalysis data. Currently, a variety of satellite products provide ice water path and cloud water path, please replace them with satellite observation data.
- 6. The wind shear is calculated using 925 and 500 hPa latitude and longitude winds, which is approximately from the ground to 5km. What does this kind of wind shear mean for a thunderstorm cloud? Wind shear in the middle and lower troposphere might be considered.
- 7. Line 134. How could a low-pressure system, which tends to bring rainy weather, cause heavy air pollution? This is very puzzling.

- 8. In section 3.2, How are clean and polluted subsets defined? When defining the polluted subset, is AOD used on the day of lightning or before the thunderstorm when there is no precipitation? Because of the significant wet deposition of precipitation, it is not reasonable to choose the aerosol on the day of lightning to define the polluted and clean subset.
- 9. Line 593. the caption does not correspond to the figure, figure b and c.
- 10. In Figure 4, there is little difference in lightning between the polluted background and the clean background between 13:00 and 18:00, and there is more lightning in the polluted background in the rest of the time. However, it is not rigorous to describe the difference between day and night in the whole paper, because the difference is only seen from the figure between the afternoon and other times.
- 11. Line 600, Which variable has a spatial resolution of 0.1°? The resolution described in the above data description is 0.5°.
- 12. In Figure 7b, in period 2 (in red), the fitting line between wind shear and lightning may not be suitable, as it should rise first and then fall. And why the apparent difference in the relationship between wind shear and lightning at the two different period (red and blue)?
- 13. Line 256, TCL?
- 14. In Figure 8b, it is generally believed that ice particles directly determine the activity of lightning. In period 1 (blue), lightning has no obvious relationship with ice water. How to explain this?
- 15. In Figure 8, the authors analyze the relationship between lightning and TCLW and TCIW. But in Figure 9, only TCLW is analyzed, not TCIW. Does lightning depend more on liquid water than ice water? The connectivity and logic here need to be improved.
- 16. Figure 9c is not well understood. Does it refer to the effect of TCLW on temperature? What is the physical mechanism by which TCLW affects surface temperature?
- 17. In Figures 11 a and b, lower TCLW corresponds to more lightning in period 1, while this relationship is reversed in period 2. Why does liquid water have opposite effects

on lightning in different periods?

- 18. In Figure 12, the authors suggest that aerosol inhibit convective activity during the day through ARIs, but in the absence of any evidence presented above, this speculation is unconvincing.
- 19. The authors attempted to explain the difference between the effects of aerosols on lightning during the day and tnight, using a schematic diagram. The authors suggest that aerosol do not exhibit radiative effects at night. Previous study (Fan et al., 2015) have suggested that in the Sichuan Basin, aerosols make the boundary layer more stable through radiation effects in the daytime, which makes the convection more vigorous at night. The viewpoint that aerosols do not affect lightning through radiation effects at nighttime cannot be accepted.

Fan, J., D. Rosenfeld, Y. Yang, C. Zhao, L. R. Leung, and Z. Li (2015), Substantial contribution of anthropogenic air pollution to catastrophic floods in Southwest China, Geophys. Res. Lett., 42, 6066–6075, doi:10.1002/2015GL064479.

20. The prominent nocturnal convective activity in the Sichuan Basin is mainly due to the relative thermal difference between the Sichuan basin and the Tibetan Plateau and the cold advection from the Tibetan Plateau to the east at night (Jin et al., 2013). The authors should consider these factors in studies to clarify the cause of the aerosol effect on nighttime lightning.

Jin X., Wu, T., Li, L. 2013. The quasi-stationary feature of nocturnal precipitation in the Sichuan Basin and the role of the Tibetan Plateau. Clim. Dyn., 41:977–994.

---

## Referee Comment (RC3)

Review on Different aerosol effects on the daytime and nocturnal cloud-to-ground lightning in the Sichuan Basin, by Wang et al.

The study has investigated the impacts of aerosols and dynamics-thermodynamics on cloud-to-ground (CG) lightning activity in the Sichuan Basin. The topic is interesting, however, there are obvious major deficiencies in the scientific analysis. Some conclusions are based on assumptions, rather than on detailed analysis of the corresponding observation results. Additional information, figures and analysis are needed to convince the reader of the validity of the conclusions. Recommendation: Major revisions.

**General comments:**

(1) The manuscript is in need of careful English language editing throughout, particularly the specific scientific term and the sentence structure. There are too many to spend time providing a full list of typos and language corrections.

(2) The data processing of polluted and clean subsets is not unambiguous (Lines 124-125). What the exact AOD range or value are used in this study? Please clarify.

(3) I still cannot understand how these two periods (Period 1 and Period2, lines 218-220) are chosen. However, these two time periods are the basis for the following analysis and discussion.

(4) How do you define the different time periods in this study? The time periods used in this paper include "nighttime (1800-600BJT)" (line 139), "midnight (2400-0100 BJT)" (line 140), "night (2300-2400 BJT)" (line 188), "midnight (2400-0300 BJT)" (line 170), "midnight (2300-0200 BJT)" (line 171), …, which is very confusing.

(5) Conclusions are based on assumptions, rather than on detailed analysis of the corresponding observation results. No statistics are presented to prove the points as follows:

(5.1) Lines 193-195: "We speculate that this may be one of the causes for the inconsistent response results of PPCG to aerosol loading in different periods."

(5.2) Lines 228-232: "Meanwhile, the relationship between aerosols and CG

lightning flashes did not show a similar nonlinear relationship at night time. We speculate that this may be due to the lack of solar radiation at night, weakening aerosol radiative effects."

(5.3) Lines 253-255: "Thus, we may infer that the thunderstorm system in the period1 is different to that in period2."

(5.4) Lines 351-353: "Therefore, it can be inferred that aerosols have different effects on lightning at different times in the study region."

(6) The results show that there are differences in the spatial distribution of CG lightning between polluted and clean subsets (Fig.5). What is the reason of this distribution? Will this influence the following analysis? Before the authors discuss the relationships between CG and aerosols at different periods, a more comprehensive discussion, related to the differences in the spatial distribution of CG lightning between polluted and clean subsets is required.

(7) Lines 240-283, Figures 7-9: How did the samples be sorted? More information about the methods should be provided. Furthermore, your conclusion seems not reliable because of the large standard deviation of each bin.

(8) Lines 256-271: The analysis seems to be completely wrong. The authors claim that "A positive relationship (r = 0.94) between them is found in Period2" (line 261, Fig. 8b). However, a negative relationship between them is shown in Fig. 8b.

(9) Figures 10-11: The authors got "564 samples" in total (line 123), however, the total number of samples in Figs. 10-11 is much larger than 564 samples, which cannot convince the reader of the validity of the conclusions.

**Technical corrections**

(5) Line 69: It should be "research" not "researches".

(6) Line 139: should be "0600 BJT" not "600 BJT".

(6) Line 256: should be "TCLW" not "TCL".

(6) Line 268: should be "TCG" not "TTCG".

(6) Line 354: should be "updraft" not "uplift".

(6) Figure 1: should be "(d)" not "(b)" (line 583).

(6) Figure 3: where is the spatial distribution of wind field you mentioned on Line 143?

(7) Figure 4: The caption does not correspond to the figure.

(8) Figure 8: The legend is completely wrong.

(9) Figure 9: should be "in (red) period1 and (blue) period2" not "in (blue) period1 and (red) period2".

(10) Figure 10-11: See comments above.

More text corrections are left for a later version.

---

## Author Comment (AC1)

Dear Editor and Referee#1,

Thank you very much for your attention and the referee's evaluation and comments on this work. Those comments are all valuable and very helpful for revising and improving our paper, as well as the important guiding significance to our research. Following are point-by-point responses to Referee #1's comments. All the line numbers mentioned in responses are referred to the manuscript with changes marked.

**Specific Comments:**

**(1)** L1-2: The title could be clearer. Perhaps "Diurnal differences in the effect of aerosols on Sichuan-Basin lightning"

**Reply**: Thank you for your comment. We have changed the title of the manuscript to "Diurnal differences in the effect of aerosols on cloud-to-ground lightning in the Sichuan Basin". Since only cloud-to-ground lightning data were analyzed in this manuscript, we thought that using "cloud-to-ground lightning" might be more appropriate in the title. (Lines: 1-2)

**(2)** L60-61: The meaning of this sentence is unclear. Are you saying that aerosol radiative effects counter microphysical effects and make it difficult to confirm the modeling results using observations or something else?

**Reply**: Thank you for your comment. What we want to express is

consistent with your comment. In some model studies, the concentration of aerosols is set very high. However, in reality, when aerosol concentration reaches such a high level, the radiation effect of aerosol often becomes very obvious and counters the microphysical effects. Therefore, it is difficult to verify these model results with observations. We have rewritten this sentence:" However, these model results are difficult to verify using observations, because the radiative effects of aerosols will offset the microphysical effects when the aerosol loading is excessively high." (Lines: 57-58)

**(3)** L90: Possible contamination by what? IC flashes? If yes, please say this.

**Reply**: Thank you for your comment. Yes, it is intracloud (IC) flashes. We have revised this in the manuscript. (Line: 90)

**(4)** L90: Since you only discuss CG flashes in this draft, you might replace all references to CG lightning with lightning – after stating once that lightning flash refers to CG lightning flash.

**Reply**: Thank you for your comment. We have revised this in the revised manuscript.

**(5)** L91: These 2 sentences are confusing. Is this what you mean?

Additionally, only the first stroke is retained if more than one stroke occurs in the next second within the first 10 km of the first stroke as two strokes that occur within 0.5 seconds are assumed to be from the same flash.

**Reply**: Thank you for your comment. What we want to express is consistent with this comment. We have rewritten these two sentences according to the comment to make them clear. The revised sentences are as follows:

"Only the first stroke is retained if more than one strokes occur in the next second with 10 km of the first stroke and two strokes that occur within 0.5 seconds are assumed to be from the same flashes (Cummins et al. 1998). In addition, it is a different flash if the polarity of the stroke is different."
(Lines: 91-93)

**(6)** L105: You mention 5 factors but discuss 7.

**Reply**: We are sorry for this mistake. We have revised this in the revised manuscript. (Line: 110)

**(7)** L123-L128: It is unclear how you obtained 564 and later 11408 samples. In addition, the reference to section 3.4 is confusing. Please rephrase this paragraph giving information such as how many grid boxes are in the region of interest? What percent of these grid boxes were excluded by the flash criterion? Also, only mention the 10 flash threshold once in the

revised paragraph.

**Reply**: Thank you for your comments. In the revised manuscript, we modified the sample processing method.

In the previous manuscript, the time of a sample includes 24 hours. It starts at 0600 BJT one day and ends at 0600 BJT the next day, as shown in Fig. 7-1.

[Figure]

Figure 7-1. Schematic diagram of sample time selection.

Then, we only retain grids with CG flashes larger than ten during the period of a sample (the blue region shown in Fig. 7-2) to make sure there are relatively strong thunderstorms in those grids (hereinafter referred to as useful grids). Only samples with useful grids will be retained. Based on this rule, we finally got 564 samples during the whole study period.

[Figure]

Figure 7-2. Black lines frame the study region. The blue region is grids with CG flashes larger than ten during the period of a sample. The spatial resolution of these grids is 0.5°×0.5°.

The AOD in a sample was calculated from the hourly averaged AOD of these grids as follows:

$$AOD_{Sample} = \frac{\sum_{k=1}^{24} AOD_{Grid,k}}{24 \times n_{Grid}}$$

The $AOD_{Sample}$ is the AOD value of a sample. The $AOD_{Grid,k}$ is the AOD value in k hour of a useful grid. The $n_{Grid}$ is the number of useful grids in a sample.

This method has some drawbacks. It did not take into account the wet deposition of aerosols by precipitation during thunderstorms. Therefore, the definition of clean and polluted subsets, as well as the analysis related to the value of AOD in the previous manuscript were not rigorous. In addition, we set a lightning threshold of ten to filter out many relatively weak lightning activities. However, these weak lightning activities should also be considered in the analysis. In the revised manuscript, we have improved the sample processing method in view of these drawbacks.

In the revised manuscript, a sample starts at 1200 BJT one day and ends at 1200 BJT the next day, as shown in Fig. 7-3 (b). In the study region, most thunderstorms from in the afternoon, at night, and the next morning (Fig. 7-3 (a)). The thunderstorms in the morning may be associated with intense thunderstorms at night. Therefore, noon is a relatively appropriate cut-off point for the sample period. The thunderstorm is weakest at noon, and the

impact of precipitation on aerosols is relatively weak. Therefore, we selected the averaged AOD of the useful grids on the first hour (between 1200 BJT and 1300 BJT) of a sample period to represent the $AOD_{Sample}$. In addition, we limited the number of grids with CG lightning flashes to less than 10% of the total grids (7 grids) in each of the six hours before the start of a sample. This is to ensure that thunderstorm has been weak for a period of time before the start of a sample to reduce the possible impact of thunderstorm precipitation on aerosol loading.

[Figure]

Figure 7-3. (a) The diurnal variation of CG lightning flashes during the study period. $Num_{Grids}$: number of grids with CG lightning flashes in each hour. $Num_{Total}$: number of grids (70) in the entire study region. (b) Schematic diagram of sample time selection.

It should be noted that the definition of useful grids has been changed to those grids with at least one CG lightning flash during a sample period. This change allowed some grids with relatively weak thunderstorms to include in the analysis. Finally, the AOD in a sample is calculated as

follows:

$$AOD_{Sample} = \frac{\sum_{k=1}^{n_{Grid}} AOD_k}{n_{Grid}}$$

The $AOD_{Sample}$ is the AOD value of a sample. The $AOD_k$ is the AOD value in the first hour of a useful grid. The $n_{Grid}$ is the number of useful grids in a sample. The samples with AOD larger than 0.8 were removed. Finally, we got 532 samples. The definition of the clean and polluted subsets is the same as the method used in the previous manuscript. All samples are sorted according to $AOD_{Sample}$ and divided into three equal sample subsets where the top third of the AOD range is labelled as polluted, and the bottom third is labelled as clean. The distribution of samples' AOD and the AOD range of clean and polluted are shown in Fig. 7-4.

[Figure]

Figure 7-4. The probability density function of ranked AOD of 532 samples. Black solid lines denote accumulated occurrence frequencies for the AOD. Red lines show the top and bottom terciles.

In section 3.4 (in raw manuscript), we aim to discuss the joint effects of aerosols and dynamics-thermodynamics factors. The analysis method we used needs enough samples. Therefore, we take each useful grid in a

sample as a new sample, thus obtaining 11408 new samples. However, this method is not rigorous, so we abandoned it in the revised manuscript. We adjust the content in section 3.4 and the analysis in section 3.4 is still based on the 532 samples used in the above content. (Lines: 138-155)

**(8)** L130: The rectangular study region shown in Figure 1a doesn't match any of the other study regions shown in the paper. Perhaps remove.

**Reply**: Thank you for your comment. We have removed this subgraph and redrawn this figure. (Line: 546)

[Figure]

Figure 8-1. (a) The terrain of the study region is on a $0.02°×0.02°$ grid. Spatial distributions of (b) aerosol optical depth (AOD) and (c) CG lightning density (flashes hour$^{-1}$ km$^{-2}$) at a spatial resolution of $0.5°×0.5°$ for the period 2010–2018 including the summer months (June, July, and August). The black lines in (a–c) outline the specific area investigated in this study.

**(9)** L143: Figure 3 does not show the wind field.

**Reply**: We are sorry for this mistake. We did not draw the wind field in this figure. We have revised the incorrect content in the manuscript.

**(10)** L181: Why do we care about hourly variations in the percentage of

**Reply**: Thank you for your comment. The space charge distribution of thunderstorms plays a crucial role in determining the polarity of lightning (Zhao et al., 2015). The space charge distribution of a thunderstorm is tightly correlated with the ice particle distribution (size, number) of the thunderstorm. By acting as CCN and IN, aerosols can affect ice particle size and number and thus affect the lightning flash polarity. Therefore, changing aerosols loading in a thunderstorm may affect its PPCG.

Some observational studies have investigated the relationship between aerosols and PPCG (Murray et al. 2000; Lang et al. 2006; Naccarato et al. 2003; Kar et al. 2014; Tan et al, 2016). Both positive and negative relationships between aerosol loading and PPCG were reported. However, the effect of aerosols on PPCG is still far from understand and related observational researches are few. Therefore, in the previous manuscript, we also analyzed the diurnal variation of the relationship between aerosol loading and PPCG.

After we change the method of processing samples (as described in the reply to comment 7), the results are not obvious (as shown in Fig. 10-1). Therefore, in the revised manuscript, we removed this part of the analysis and focused on the impact of aerosols on the lightning quantity in the Sichuan Basin. However, we believe this is an interesting subject worthy of further study. In future research, we will adopt more appropriate

methods to conduct a more comprehensive study about this

[Figure]

Figure 10-1. (a) The diurnal variation of the percentage of positive polarity CG lightning flashes (PPCG) in clean and polluted subsets. (b) The difference in the PPCG between polluted and clean subsets.

**(11)** L203-204: Warm (cold) colours in the figure mean more (less) … subset. Consider moving this sentence to the caption for Figure 5.

**Reply**: Thank you for your comment. We have moved this sentence to the caption for Figure 6 in the revised manuscript.

**(12)** L200-216: Discussion of Figure 5: Could you calculate and show the percent of 0.1 x 0.1 degree grid boxes where the change is positive and also give the mean change (amount and percent) for each of the 8 regions.

**Reply**: Thank you for your comment. We have redrawn this figure and added this information to it (Fig. 12-1). Related discussion is also added in

the manuscript.

[Figure]

Figure 12-1. (a-d, i-l) Diurnal changes of total CG lightning flash differences (unit: flashes hour$^{-1}$) between polluted and clean subset (polluted−clean) during the study period with an interval of 3 hours (BJT). Black lines represent the 1500m contour lines. The spatial resolution is 0.5°×0.5°. Warm (cold) colours in the figure mean more (less) total CG lightning flashes in the polluted subset. Plus signs denote those grids with relatively large lightning flashes difference (the absolute value of lightning flashes difference ranks in the top third). (e-h, m-p) Histograms of the differences (red: positive, blue: negative) between lightning flashes in the polluted and clean subsets. The percentages of grids with the positive (negative) difference in the total grids, the total change of lightning flashes, and its percentage are also given.

**(13)** L219: You repeatedly refer to Period1 and Period2 over the next several pages. It might be better to replace these terms with morning and middle-of-the-night or something meaningful.

**Reply**: Thank you for your comment. We use "afternoon" and "night" to represent these two periods in the revised manuscript.

**(14)** L233-234: It is unclear what you mean by this sentence. Are you saying that you see a 0.3 threshold during the day in this study consistent with other studies? If yes, state this more clearly.

**Reply**: Thank you for your comment. We have rewritten the relevant content in the revised manuscript and stated it more clearly.

**(15)** L253-255: Are there any scientific studies of convection in the Sichuan Basin that support this inference? If yes, please reference them.

**Reply**: Thank you for your comment. We are sorry that we have not found relevant studies that directly point out this difference in convective activities in Sichuan Basin. We revised the relevant speculation in the article. It is not rigorous for that inference.

**(16)** L256: TCL is negatively ï□ Be clear as to whether you mean TCLW or TCIW.

**Reply**: We are sorry for this mistake. We have revised it in the revised manuscript.

**(17)** L272-276: Check the captions in Figure 9 and make sure TCLW and TCIW are used appropriately. They probably all should be labeled TCLW.

**Reply**: Thank you for your comment. We have redrawn this figure and

revised these mistakes in it.

**(18)** L296: Rather than stating that more CG flashes are found it would be more interesting if you could give a percent increase range by dividing values from a subset of the bins.

**Reply**: Thank you for your comment. Due to the modification of the data processing method of the article, we have made some adjustments to the content of this section. This content was removed.

**(19)** L305: Rather than "more marked" cite a percent change. This should be done throughout L296-305.

**Reply**: Thank you for your comment. We have rewritten this part of the content in the revised manuscript.

**(20)** L357: Hopefully, you can support this inference by other studies.

**Reply**: Thank you for your comment. We have revised the relevant speculation in the article. It is not rigorous for that inference.

**(21)** L591-595: Figure 4 caption does not match Figure 4.

**Reply**: We are sorry for this mistake. We have revised it in the revised manuscript.

**(22)** Figure 5: Be sure to use BJT consistently as opposed to BJ.

**Reply**: Thank you for your comment. We have revised these mistakes in the revised manuscript.

**(23)** L604: Here and elsewhere consider replacing "the error was calculated" with "the uncertainty was calculated".

**Reply**: Thank you for your comment. We have revised this in all relevant figure captions.

**(24)** Figures 10 and 11: It might make more sense to show the flash counts with the numbers rather than the number of samples in the cell. This would emphasize your main points and give the reader more interesting numbers to play with.

**Reply**: Thank you for your comment. In section 3.4 (in raw manuscript), we aim to discuss the joint effects of aerosols and dynamics-thermodynamics factors. The analysis method we used needs enough samples. Therefore, we take each useful grid in a sample as a new sample, thus obtaining 11408 new samples. However, this method is not rigorous, so we abandoned it in the revised manuscript. We adjust the content in section 3.4 and the analysis in section 3.4 is still based on the 532 samples processed by the improved method (as described in the reply to comment

7). However, this suggestion is very meaningful, and we will pay attention to it in similar drawings in the future work.

**Technical Corrections:**

**Reply**: Thank you for your technical corrections. We have revised all these errors in the revised manuscript.

**References**:

Kar, S.K., and Liou, Y.: Enhancement of cloud-to-ground lightning activity over Taipei, Taiwan in relation to urbanization, Atmos. Res., 147, 111–120, https://doi.org/10.1016/J.ATMOSRES.2014.05.017, 2014.

Lang, T.J., and Rutledge, S.A.: Cloud-to-ground lightning downwind of the 2002 Hayman forest fire in Colorado, Geophys. Res. Lett., 33, L03804, https://doi.org/10.1029/2005GL024608, 2006.

Murray, N.D., Orville, R.E., and Huffines, G.R.: Effect of pollution from Central American fires on cloud-to-ground lightning in May 1998, Geophys. Res. Lett., 27, 2249–2252, https://doi.org/10.1029/2000GL011656, 2000.

Zhao, P., Yin, Y., and Xiao, H.: The effects of aerosol on development of thunderstorm electrification: A numerical study, Atmos. Res., 153, 376–391, https://doi.org/10.1016/J.ATMOSRES.2014.09.011, 2015.

Naccarato, K.P., Pinto, O., and Pinto, I.R.: Evidence of thermal and aerosol

effects on the cloud-to-ground lightning density and polarity over large urban areas of Southeastern Brazil, Geophys. Res. Lett., 30, 1674, https://doi.org/10.1029/2003GL017496, 2003.

Tan, Y., Peng, L.K., Shi, Z., and Haiqin, C.: Lightning flash density in relation to aerosol over Nanjing (China), Atmos. Res., 174, 1–8, https://doi.org/10.1016/J.ATMOSRES.2016.01.009, 2016.

---

## Author Comment (AC2)

Dear Editor and Referee#2,

Thank you very much for your attention and the referee's evaluation and comments on this work. Those comments are all valuable and very helpful for revising and improving our paper, as well as the important guiding significance to our research. Following are point-by-point responses to Referee #2's comments. All the line numbers mentioned in responses are referred to the manuscript with changes marked.

(1) There is no outstanding innovation in this paper, including analysis methods and conclusions. The authors are requested to extract the innovative points of this paper in the introduction and emphasize the innovative points in the conclusion, on the premise of adjusting the research content.

**Reply**: Thank you for your comment. We have adjusted the content of the manuscript. The main purpose of this study is to investigate the difference between the effects of aerosols on lightning during the day and at night. Previous studies show that the relationship between aerosol and lightning is very complicated. Aerosols may promote or inhibit lightning activities, or have no obvious impact on lightning activities. One of the reasons for this phenomenon is that aerosol radiation inhibition and microphysical promotion are often combined, and environmental factors should also be considered. Some studies based on hourly data reveal that the aerosolinhibited effect on lightning weakens after sunset (Guo et al. 2016). The current study aims to investigate the difference in the effects of aerosols on lightning under conditions with and without solar radiation. We have emphasized these in the introduction and conclusion part in the revised manuscript. (Lines: 68-73, 314-350)

(2) Line 81. In this paper, the data with 0.5° spatial resolution are selected to discuss the relationship between aerosol and lightning activity in the Sichuan Basin. Is it statistically significant to analyze the data with such rough resolution in such a limited space?

**Reply**: Thank you for your comment. The minimum resolution of multidata limits the data resolution used in the final analysis of this paper. But we have used the data of a long time series to obtain enough samples. For the topic to be analyzed in this paper, these data are relatively sufficient. Although the study region in this paper has a limited spatial scope, the lightning activity in this area is significantly larger than that in the surrounding areas at night (Xia et al. 2015). The aerosol value in this area is also significantly higher than that in the surrounding areas. However, the coarse resolution of these data will also bring some unavoidable problems. There will be some deficiencies in the interpretation of some phenomena. In future work, we will look for and use higher-quality data to further explore the possible impact of aerosols on lightning activities in the study region and its surrounding region.

(3) It is suggested to use the satellite lightning data to verify the groundbased lightning data used in this paper.

**Reply**: Thank you for your comment. We compared the data of groundbased CG lightning and satellite-based lightning density (from LIS), including spatial distribution and diurnal variation (Fig. 3-1). Overall, the lightning data from the ground and satellite were similar. This have been added to the supplementary materials.

Figure 3-1. Spatial distribution of lightning density (flashes hour-1 km-2) from (a) ground and (b) satellite at a spatial resolution of 0.5°×0.5°. (c) Diurnal variation of lightning occurrence frequency (OF).

(4) Line 95. First, there is an obvious error. The spatial resolution of the AOD data of MERRA-2 is not  $0.5^{\circ} \times 0.5^{\circ}$ , please check and confirm. Since the resolution is not  $0.5^{\circ}$ , how to match and discuss with other data is a

major problem. Secondly, the AOD data selected in this paper are reanalysis data. In the study area, the authors did not compare with the satellite observation and ground-based observation, so it is obviously unreasonable not to verify the availability of the data.

**Reply**: Thank you for your comment. Yes, the raw spatial resolution of the AOD data of MERRA-2 should be  $0.5^{\circ} \times 0.625^{\circ}$ . We downloaded the AOD data from MERRA's official website, which provides tools to change the resolution (as shown in Fig. 4-1). Among them, we chose bilinear interpolation to process the spatial resolution of AOD to  $0.5^{\circ} \times 0.5^{\circ}$  to match with other data. We have added this process in the revised manuscript.

Figure 4-1

We compared the AOD data of MERRA and MODIS in the study area (as shown in Fig. 4-2). It can be found that the AOD data of MERRA is well correlated with the AOD data of MODIS in the study region. These have been added to the supplementary materials.

(5) Line 105. What is the time resolution of the thermodynamic and cloudrelated data selected in this paper? Please clarify. TCIW and TCLW are reanalysis data. Currently, a variety of satellite products provide ice water path and cloud water path, please replace them with satellite observation data.

**Reply**: Thank you for your comment. The time resolution of the thermodynamic and cloud-related data selected in this paper is hourly. We have clarified this in the revised manuscript. Although a large number of satellite data can provide cloud ice and liquid water path data, there are still some deficiencies in the continuity of space and time, which cannot meet the needs of this study. Therefore, this paper selects the reanalysis data to analyse.

(6) The wind shear is calculated using 925 and 500 hPa latitude and longitude winds, which is approximately from the ground to 5km. What does this kind of wind shear mean for a thunderstorm cloud? Wind shear

**in the middle and lower troposphere might be considered.**

**Reply**: Thank you for your comment. In the previous manuscript, the selection of wind shear was referred to Wang et al. (2018). It may not be suitable for the study region of this study because of the large latitude difference between the study region in this study and that of Wang et al. (2018).

In the revised manuscript, we selected wind shear in the low (850hPa to 700hPa, about 1.5km~3km) and middle (500hPa to 400hPa, about 5km~7km) troposphere, respectively. The relationship between the wind shear and CG lightning flashes is shown in Figs. 6-1 and 6-2. In period1, CG lightning flashes decrease with the increase of wind shear in the low and middle troposphere (Fig. 6-1a and Fig. 6-2a). In period2, a similar relationship was found between CG lightning flashes and wind shear in the middle troposphere (Fig. 6-2b) but a reversed relationship was found between CG lightning flashes and wind shear in the low for fig. 6-2b).

---

## Author Comment (AC3)

Dear Editor and Referee#3,

Thank you very much for your attention and the referee's evaluation and comments on this work. Those comments are all valuable and very helpful for revising and improving our paper, as well as the important guiding significance to our research. Following are point-by-point responses to Referee #3's comments. All the line numbers mentioned in responses are referred to the manuscript with changes marked.

**(1)** The manuscript is in need of careful English language editing throughout, particularly the specific scientific term and the sentence structure. There are too many to spend time providing a full list of typos and language corrections.

**Reply**: Thank you for your comment. We are sorry for these language mistakes in this manuscript. We have revised most of the content in the article and carefully checked the language to reduce these errors.

**(2)** The data processing of polluted and clean subsets is not unambiguous (Lines 124-125). What the exact AOD range or value are used in this study? Please clarify.

**Reply**: Thank you for your comment. We have rewritten this part and added a figure to illustrate the distribution of samples' AOD value and the AOD range of clean and polluted subsets (Fig. 2-1). (Lines: 138-155)

[Figure]

Figure 2-1. The probability density function of ranked AOD of 532 samples. Black solid lines denote accumulated occurrence frequencies for the AOD. Red lines show the top and bottom terciles.

**(3)** I still cannot understand how these two periods (Period 1 and Period2, lines 218-220) are chosen. However, these two time periods are the basis for the following analysis and discussion.

**Reply**: Thank you for your comments. In this manuscript, we aim to investigate the diurnal differences in the effect of aerosols on lightning in the Sichuan Basin. By comparing the diurnal variation of CG lighting flashes under clean and polluted subsets, we found that the difference in the response of CG lightning flashes to aerosols mainly occurred between the afternoon and other times (night and morning). Little difference between the CG lightning flashes was found between the clean and polluted subset, while at other times (especially around midnight), the CG lightning flashes in the polluted subset were markedly greater than that in the clean subset. Therefore, we selected two time periods in the afternoon and night respectively in the following content to investigate the

relationship between CG lightning flashes and aerosols, thermodynamics-dynamics factors and cloud-related parameters.

**(4)** How do you define the different time periods in this study? The time periods used in this paper include "nighttime (1800-600BJT)" (line 139), "midnight (2400-0100 BJT)" (line 140), "night (2300-2400 BJT)" (line 188), "midnight (2400-0300 BJT)" (line 170), "midnight (2300-0200 BJT)" (line 171), …, which is very confusing.

**Reply**: Thank you for your comment. We are sorry for these unclear descriptions in the manuscript. We have revised these descriptions in the revised manuscript. We direct describe the different time periods using numbers like "1200-1800 BJT".

**(5)** Conclusions are based on assumptions, rather than on detailed analysis of the corresponding observation results. No statistics are presented to prove the points as follows:

(5.1) Lines 193-195: "We speculate that this may be one of the causes for the inconsistent response results of PPCG to aerosol loading in different periods."

(5.2) Lines 228-232: "Meanwhile, the relationship between aerosols and CG lightning flashes did not show a similar nonlinear relationship at night time. We speculate that this may be due to the lack of solar radiation at

night, weakening aerosol radiative effects."

(5.3) Lines 253-255: "Thus, we may infer that the thunderstorm system in the period1 is different to that in period2."

(5.4) Lines 351-353: "Therefore, it can be inferred that aerosols have different effects on lightning at different times in the study region."

**Reply**: Thank you for your comment. Given these comments, we have revised the manuscript's content. The specific modifications are as follows:

**Reply to 5.1:** In the previous manuscript, we investigated the diurnal variation of PPCG under polluted and clean conditions. The results showed that there were some differences in PPCG's response to AOD at different times of the day, but they were not obvious. After we improved the sample processing method (as described in the reply to comment 9), the difference in PPCG response to AOD became less obvious (as shown in Fig. 5-1). Therefore, we decided to remove this part of the PPCG analysis in the revised version. We will mainly focus on investigating the difference between the daytime and nighttime effects of aerosols on lightning frequency in the study area, which is also the main purpose of this study at the beginning. Some studies have reported the relationship between aerosols and PPCG, but different results have been found in different regions (i.e., positive correlation: Tan et al. 2016; Murray et al. 2000; negative correlation: Kar et al. 2014; Naccarato et al. 2003). This is an interesting subject worthy of further study. In future research, we will adopt

more appropriate methods to conduct a more comprehensive study about this.

[Figure]

Figure 5-1. (a) The diurnal variation of the percentage of positive polarity CG lightning flashes (PPCG) in clean and polluted subsets. (b) The difference in the PPCG between polluted and clean subsets

**Reply to 5.2 and 5.4**: Firstly, in the previous manuscript, our analysis mainly compared the differences in the relationship between aerosols and lightning flashes in the afternoon and part of the night. The descriptions such as "Therefore, it can be inferred that aerosols have different effects on lightning at different times in the study region" in the article are not rigorous. We have revised these descriptions in the manuscript.

The reason why we chose these two time periods is that we found that the difference in the relationships between aerosols and lightning flashes in these two time periods was the most obvious. In the afternoon (1200-1800

BJT), the lightning flashes have little difference in polluted and clean subsets, while at night (2300-0500 BJT), the lightning flashes have the largest difference in polluted and clean subsets (as shown in Fig. 5-2).

[Figure]

Figure 5-2. (a) The diurnal variations in CG lightning flashes under clean and polluted subsets over the study region during the summer (June, July, and August) season of 2010-2018. (b) The histogram of the difference in CG lightning flashes between the polluted and clean subsets

Further analysis shows that the relationships between AOD and lightning flashes in these two periods (afternoon: 1200-1800 BJT, night: 2300-0500 BJT) are apparent different (as shown in Fig. 5-3). The AOD and lightning flashes show different nonlinear relationships in the afternoon and night. The lightning flashes first increase with the increase of AOD and then decrease when AOD exceed about 0.3. At night, the lightning flashes also first increase with the increase of AOD but change little when AOD exceed about 0.3. Some studies also reported a similar nonlinear relationship between aerosol loading and lightning flashes and found a similar tipping

point of AOD (Wang et al. 2018: ~0.3, Altaratz et al. 2010, Koren et al. 2008: ~0.25). The microphysical effect of aerosols increases with the increase of aerosols loading, but when the aerosol concentration exceeds a threshold value, the microphysical effect of aerosols will reach saturation. On the contrary, the direct effect of aerosols is weak when the aerosol concentration is relatively low, and will gradually become stronger with the increase of the aerosol concentration (Rosenfeld et al. 2008). The relative intensity of these two effects of aerosols changes with the aerosol concentration, which may result in the nonlinear relationship between aerosols and lightning frequency.

[Figure]

Figure 5-3. Relationships between lightning flashes and AOD in (a) afternoon (1200–1800 BJT) and (b) night (2300–0500 BJT). Note that samples are first sorted by AOD and then samples with similar AOD were averaged to create the presented scatter plot.

The max number of samples in each bin is equal to or less than 20. The difference between the maximum and minimum AOD values of samples in each bin is equal to or less than 0.05. An estimation of the error was calculated from the standard deviation of each bin divided by the square root of the number of data points in the bin. Linear-fit lines, Pearson correlation coefficients (r), and significant level (p) are also shown.

In this study, when the aerosol loading is relatively low (AOD<~0.3), the aerosols both positively correlated with the lightning flashes in the two time periods. The biggest difference in the influence of aerosols on lightning flashes in these two time periods occurs when the concentration of aerosols is relatively high (AOD>~0.3). We speculate that this may be caused by the different roles of solar radiation in the afternoon and at night in the study region, as well as special topographic and meteorological conditions. To prove this, we added the analysis of the relationship between 2m temperature and AOD, cloud-related factors (as shown in Fig. 5-4). When the aerosol loading is relatively high, the aerosol layer above the surface will reduce the solar radiation that reaches the surface by absorbing or scattering the solar radiation and thus reducing the surface temperature (Fig. 5-4a). This effect will disappear at night because of the absence of solar radiation during this time period. Therefore, no significant relationship can be found between AOD and T at night (Fig. 5-4e). The increase in cloud liquid water will lead to thicker and larger clouds and prevent solar radiation from reaching the ground. Meanwhile, too much cloud liquid water may promote the development of the warm-rain process and further reduce the T. On the other hand, the increase in T is also

conducive to the rise of water vapour which is conducive to an increase in cloud liquid water. In the afternoon, when the TCLW is less than about 0.1 kg m$^{-2}$, the relationship between T and TCLW is unclear. However, at night, when the TCLW is less than about 0.1 kg m$^{-2}$, TLCW is positively correlated with T. This shows that when the TCLW is relatively small (<~0.1 kg m$^{-2}$) and the precipitation process has not yet formed, in the afternoon, the increase of cloud water content by T and the decrease of T by TCLW through blocking solar radiation cancel each other, resulting in an insignificant relationship between T and TCLW. At night, due to the absence of solar radiation, the effect of TCLW reducing T by blocking solar radiation disappears, and the effect of T on the increase of TCLW is dominant, resulting in a positive correlation between them. When the TCLW is relatively large (>~0.1 kg m$^{-2}$), no matter in the afternoon or at night, the warm-rain process is promoted, and the evaporation of precipitation on the surface reduces the T. In the afternoon, the relationship between TCIW and T is similar to that between TCLW and T (Fig. 5-4c). However, at night, no obvious relationship was found between TCIW and T (Fig. 5-4g). This may be due to the fact that the TCIW is more related to the strength of the updraft. The T at night is not the main factor affecting the convection intensity, but the conversion process from TCIW to precipitation is more complex, so the TCIW has a weaker effect on the reduction of the T. In addition, increase aerosol loading will produce more

but smaller cloud droplets that inhibit the warm rain process and may lead

to an increase in cloud liquid water (Fig. 5-4d and h). Therefore, the reason

for the negative relationship between AOD and T in the afternoon may also

include the effect of aerosol on cloud water content. In summary, in the

afternoon, excessive aerosols will reduce the T through its direct radiative

effects and microphysical effects. At night, such inhibit effects on T are

reduced. The reduced T leads to the increase of atmospheric stability and

thus inhibits lightning activity. This may explain the difference in the

relationships between AOD and lightning flashes in the afternoon and night.

[Figure]

Figure 5-4. Relationships in (a and e) AOD-T, (b and f) TCLW-T, (c and g) TCIW and (d and h) AOD-TCLW in the afternoon (1200–1800 BJT) and night (2300–0500 BJT). Note that samples are first sorted by (a, e, d, and h) AOD, (b, f) TCLW, and (c, g) TCIW, and then samples with similar (a, e, d, and h) AOD, (b, f) TCLW, and (c, g) TCIW were averaged to create the presented scatter plot. The max number of samples in each bin is equal to or less than 20. The difference between the maximum and minimum AOD, TCLW, and TCIW values of samples in each bin is equal to or less than 0.05, 0.05 and 0.01. An estimation of the error was calculated from the standard deviation of each bin divided by the square root of the number of data points in the bin. Smoothing spline-fit curves, Pearson correlation coefficients (r), and significant level (p) are also shown in each panel.

**Reply to 5.3**: This inference is not rigorous. Based on the diurnal change

of the spatial distribution of lightning flashes (as shown in Fig. 5-5), the

spatial distribution of lightning has obvious differences in different time

periods. Compared with the afternoon, nighttime lightning mainly occurred in the south and west part of the study region. This regional difference may also be the reason why the relationship between vertical wind shear and lightning flashes is different in the afternoon and at night. We have revised the relevant statements in the manuscript.

[Figure]

Figure 5-5. Diurnal cycle of total CG lightning flashes (unit: flashes hour$^{-1}$) on a 0.5°×0.5° grid with an interval of 3 hours (BJT) for 2010–2018 including the summer months (June, July, and August). The black lines represent the 1500m contour lines.

**(6)** The results show that there are differences in the spatial distribution of CG lightning between polluted and clean subsets (Fig.5). What is the reason of this distribution? Will this influence the following analysis? Before the authors discuss the relationships between CG and aerosols at different periods, a more comprehensive discussion, related to the differences in the spatial distribution of CG lightning between polluted and clean subsets is required.

**Reply**: Thank you for your comment. We have redrawn this figure and added more information (as shown in Fig. 6-1). Between 2100 BJT and

0900 BJT, the difference in lightning flashes under polluted and clean subsets is obvious, but it is not obvious for the rest of the time (the change of lightning flashes of most grid points is less than 1). During this period, the most obvious differences were concentrated in the south and northwest parts of the study region (Figure 6-1d, I, j, and k).

[Figure]

Figure 6-1. (a-d, i-l) Diurnal changes of total CG lightning flash differences (unit: flashes hour$^{-1}$) between polluted and clean subset (polluted−clean) during the study period with an interval of 3 hours (BJT). Black lines represent the 1500m contour lines. The spatial resolution is 0.5°×0.5°. The colour in a grid represents the value of lightning flashes change in the grid. Plus signs denote those grids with relatively large lightning flashes difference (the absolute value of lightning flashes difference ranks in the top third). (e-h, m-p) Histograms of the differences (red: positive, blue: negative) between lightning flashes in the polluted and clean subsets. The percentages of grids with the positive (negative) difference in the total grids, the total change of lightning flashes, and its percentage are also given.

Fig. 6-2 and 6-3 show the diurnal cycle of lightning flashes in polluted and clean subsets, respectively. In general, the spatial distribution of lightning flashes under polluted and clean subsets is similar, especially between 1800

and 0600 BJT. We speculate that the spatial distribution of lightning flashes in the study region is mainly controlled by terrain and meteorological conditions, and aerosol may have little impact on its spatial distribution. The difference brought by aerosols may be mainly reflected in the time difference. In addition, this difference in the spatial distribution of lightning flashes needs to be considered in the following analysis. This may be the reason for the difference in lightning flashes and other factors (such as vertical wind shear) between afternoon and night.

[Figure]

Figure 6-2. Diurnal cycle of lightning flashes in polluted subset on a 0.5°×0.5° grid with an interval of 3 hours (BJT) for 2010–2018 including the summer months (June, July, and August). The black lines represent the 1500m contour lines.

[Figure]

Figure 6-3. Same as in Fig. 6-2, but for lightning flashes in a clean subset.

**(7)** Lines 240-283, Figures 7-9: How did the samples be sorted? More information about the methods should be provided. Furthermore, your conclusion seems not reliable because of the large standard deviation of each bin.

**Reply**: Thank you for your comment. In the previous manuscript, when creating the scatter plot between lightning flashes and other factors (referred to as x), the lightning data were first sorted as a function of x and then every 20 points were averaged. This method does not control the range of x in each bin, resulting in a large standard deviation in some bins. We have improved this method in the revised manuscript. The samples were first sorted as a function of x and then samples with similar x were averaged. The max number of samples in each bin is equal to or less than 20. The difference between the maximum and minimum x of samples in each bin is limited to a fixed range. This information has been added to the figure

title. With the improved method, the standard deviation in each figure is lower than the previous results.

**(8)** Lines 256-271: The analysis seems to be completely wrong. The authors claim that "A positive relationship ($r = 0.94$) between them is found in Period2" (line 261, Fig. 8b). However, a negative relationship between them is shown in Fig. 8b.

**Reply**: We are sorry for this mistake. We have checked and revised this wrong analysis in the revised manuscript.

**(9)** Figures 10-11: The authors got "564 samples" in total (line 123), however, the total number of samples in Figs. 10-11 is much larger than 564 samples, which cannot convince the reader of the validity of the conclusions.

**Reply**: Thank you for your comments. In the revised manuscript, we modified the sample processing method.

In the previous manuscript, the time of a sample includes 24 hours. It starts at 0600 BJT one day and ends at 0600 BJT the next day, as shown in Fig. 9-1.

[Figure]

A sample

2010.06.25.0600          2010.06.25.2400   2010.06.26.0600

Time (BJT)

Figure 9-1. Schematic diagram of sample time selection.

Then, we only retain grids with CG flashes larger than ten during the period of a sample (the blue region shown in Fig. 9-2) to make sure there are relatively strong thunderstorms in those grids (hereinafter referred to as useful grids). Only samples with useful grids will be retained. Based on this rule, we finally got 564 samples during the whole study period.

[Figure]

Figure 9-2. Black lines frame the study region. The blue region is grids with CG flashes larger than ten during the period of a sample. The spatial resolution of these grids is 0.5°×0.5°.

The AOD in a sample was calculated from the hourly averaged AOD of these grids as follows:

$$AOD_{Sample} = \frac{\sum_{k=1}^{24} AOD_{Grid,k}}{24 \times n_{Grid}}$$

The $AOD_{Sample}$ is the AOD value of a sample. The $AOD_{Grid,k}$ is the AOD value in k hour of a useful grid. The $n_{Grid}$ is the number of useful grids in a

sample.

This method has some drawbacks. It did not take into account the wet deposition of aerosols by precipitation during thunderstorms. Therefore, the definition of clean and polluted subsets, as well as the analysis related to the value of AOD in the previous manuscript were not rigorous. In addition, we set a lightning threshold of ten to filter out many relatively weak lightning activities. However, these weak lightning activities should also be considered in the analysis. In the revised manuscript, we have improved the sample processing method in view of these drawbacks.

In the revised manuscript, a sample starts at 1200 BJT one day and ends at 1200 BJT the next day, as shown in Fig. 9-3 (b). In the study region, most thunderstorms from in the afternoon, at night, and the next morning (Fig. 9-3 (a)). The thunderstorms in the morning may be associated with intense thunderstorms at night. Therefore, noon is a relatively appropriate cut-off point for the sample period. The thunderstorm is weakest at noon, and the impact of precipitation on aerosols is relatively weak. Therefore, we selected the averaged AOD of the useful grids on the first hour (between 1200 BJT and 1300 BJT) of a sample period to represent the $AOD_{Sample}$. In addition, we limited the number of grids with CG lightning flashes to less than 10% of the total grids (7 grids) in each of the six hours before the start of a sample. This is to ensure that thunderstorm has been weak for a period of time before the start of a sample to reduce the possible impact of

thunderstorm precipitation on aerosol loading.

[Figure]

Figure 9-3. (a) The diurnal variation of CG lightning flashes during the study period. Num$_{Grids}$: number of grids with CG lightning flashes in each hour. Num$_{Total}$: number of grids (70) in the entire study region. (b) Schematic diagram of sample time selection.

It should be noted that the definition of useful grids has been changed to those grids with at least one CG lightning flash during a sample period. This change allowed some grids with relatively weak thunderstorms to include in the analysis. Finally, the AOD in a sample is calculated as follows:

$$AOD_{Sample} = \frac{\sum_{k=1}^{n_{Grid}} AOD_k}{n_{Grid}}$$

The AOD$_{Sample}$ is the AOD value of a sample. The AOD$_k$ is the AOD value in the first hour of a useful grid. The n$_{Grid}$ is the number of useful grids in a sample. Finally, we got 532 samples. The definition of the clean and polluted subsets is the same as the method used in the previous manuscript. All samples are sorted according to AOD$_{Sample}$ and divided into three equal

sample subsets where the top third of the AOD range is labelled as polluted, and the bottom third is labelled as clean. The distribution of samples' AOD and the AOD range of clean and polluted are shown in Fig. 9-4.

[Figure]

Figure 9-4. The probability density function of ranked AOD of 532 samples. Black solid lines denote accumulated occurrence frequencies for the AOD. Red lines show the top and bottom terciles.

In section 3.4 (in raw manuscript), we aim to discuss the joint effects of aerosols and dynamics-thermodynamics factors. The analysis method we used needs enough samples. Therefore, we take each useful grid in a sample as a new sample, thus obtaining 11408 new samples. However, this method is not rigorous, so we abandoned it in the revised manuscript. We adjust the content in section 3.4 and the analysis in section 3.4 is still based on the 532 samples used in the above content.

**Technical corrections:**

**Reply**: Thank you for your technical corrections. We have revised all these errors in the revised manuscript.

**References**:

Altaratz, O., Koren, I., Yair, Y.Y., and Price, C.G.: Lightning response to smoke from Amazonian fires, Geophys. Res. Lett., 37, L07801, https://doi.org/10.1029/2010GL042679, 2010.

Kar, S.K., and Liou, Y.: Enhancement of cloud-to-ground lightning activity over Taipei, Taiwan in relation to urbanization, Atmos. Res., 147, 111–120, https://doi.org/10.1016/J.ATMOSRES.2014.05.017, 2014.

Koren, I., Martins, J.V., Remer, L.A., and Afargan, H.: Smoke Invigoration Versus Inhibition of Clouds over the Amazon, Science, 321, 946–949, https://doi.org/10.1126/science.1159185, 2008.

Tan, Y., Peng, L.K., Shi, Z., and Haiqin, C.: Lightning flash density in relation to aerosol over Nanjing (China), Atmos. Res., 174, 1–8, https://doi.org/10.1016/J.ATMOSRES.2016.01.009, 2016.

Murray, N.D., Orville, R.E., and Huffines, G.R.: Effect of pollution from Central American fires on cloud-to-ground lightning in May 1998, Geophys. Res. Lett., 27, 2249–2252, https://doi.org/10.1029/2000GL011656, 2000.

Naccarato, K.P., Pinto, O., and Pinto, I.R.: Evidence of thermal and aerosol effects on the cloud-to-ground lightning density and polarity over large urban areas of Southeastern Brazil, Geophys. Res. Lett., 30, 1674, https://doi.org/10.1029/2003GL017496, 2003.

Rosenfeld, D., Lohmann, U., Raga, G.B., O'Dowd, C.D., Kulmala, M., Fuzzi, S., Reissell, A., and Andreae, M.O.: Flood or Drought: How Do Aerosols Affect Precipitation? Science, 321, 1309–1313, https://doi.org/10.1126/science.1160606, 2008.

Saunders, C.P.: Charge Separation Mechanisms in Clouds, Space. Sci. Rev., 137, 335–353, https://doi.org/10.1007/S11214-008-9345-0, 2008.

Wang, Q., Li, Z., Guo, J., Zhao, C., and Cribb, M.: The climate impact of aerosols on the lightning flash rate: is it detectable from long-term measurements? Atmos. Chem. Phys., 18, 12797–12816, https://doi.org/10.5194/ACP-18-12797-2018, 2018.

---

## Referee Report (RR1)

Review on Different aerosol effects on the daytime and nocturnal cloud-to-ground lightning in the Sichuan Basin, by Wang et al.
January 2023.

This version of the paper is a large improvement on the previous version. There are now physical explanations for the results, and the paper is now much more informative than it was before. I have a few comments:

(1) Lines 12-17: The authors first claimed that "the difference in lightning flashes between the clean and polluted subset was not obvious in the afternoon", however, the authors continued to say that "increasing AOD will lead to an increase in lightning flashes in the afternoon and night". These two statements are contradictory. The abstract is very important for readers, and the authors are expected to write it as clearly as possible.

(2) Lines 232-235: Why are the correlation coefficients between lightning flashes and temperature the same (R = 0.68) in the afternoon and evening? The relationships in Figs. 8a and 8e are not linear relationship, but here it is called a positive relationship? Namely, "Positive relationships can be found in lightning flashes and T both in the afternoon and at night". This description is inaccurate. Please clarify.

(3) In addition, what is the reason for the above nonlinear relationship?

(4) Lines 255-258: The authors claimed that "The CG lightning flashes increase with the increase of TCLW when the TCLW is relatively low (<~01 kg m-2), but decrease with the rise of TCLW when its value exceeds about 0.1 kg m-2". But the explanation given by the authors cannot convince me. Zhao et al. (2020) also discussed the relationship between these two and indicated that it was related to different regions in this region. Please discuss these two different explanations.

Zhao P , Li Z , Xiao H , et al. Distinct aerosol effects on cloud-to-ground lightning in the plateau and basin regions of Sichuan, Southwest China[J]. Atmospheric Chemistry and Physics, 2020, 20(21).

(5) Lines 279-284: The results show that "TCLW was negatively correlated with T in

the afternoon and at night". And the authors gave two different explanations for this phenomenon. Firstly, "The thicker and wider clouds will block more solar radiation from reaching the ground, thus reducing the surface temperature" (line 278). Secondly, "too much liquid water in the loud may promote a warm-rain process, The precipitation falling to the ground will significantly reduce the surface temperature" (lines 283-284). Please provide the references for these two explanations separately. And which factor is more important?

(6) Line 287: The authors claimed that "This may be because the ice water content in clouds is related to more factors". Please provide these factors and the references accordingly.

(7) Lines 331: In which region are these analyses conducted? Please clarify.

(8) Lines339-340: What the authors claimed "reduce the solar radiation reaching the ground" is not the "microphysical effects" of aerosols. The usage of proper nouns should be checked in the paper.

(9) I grew tired well reading the paper, which is an indication that the writing needs to be improved.

---

## Author Response (AR2)

Dear Editor and Referees,

We are very grateful for your comments on the manuscript. According to your advice, we amended the relevant part of the manuscript. Following are point-by-point responses to comments in Referee #1 Report #3 and Referee #3 Report #2. All the line numbers mentioned in responses are referred to the manuscript with changes marked.

**Referee #1 Report #3:**

**Specific Comments:**

**(1)** L148: This sentence is confusing.

"In addition, within six hours before the start of each sample, the number of grids with CG lightning flashes records in each hour is less than 10% of the entire study region. This is to ensure that thunderstorm is …"

Is this what your mean?

"In addition, days when flashes occur in more than 10% of the study region during any of the six-hours (06-12 UT) preceding the sample period are removed to ensure that thunderstorm activity is …"

**Reply**: Thank you for your comment. Yes, this is what we want to express. We have revised this sentence according to this comment. (Lines: 157-159)

(2) L184: You could argue that there is only one peak on clean days too with the peak lasting from 1700-0100 BJT. Do you believe the decrease in

flashes between the two small peaks is significant or noise?

**Reply**: Thank you for your comment. The decrease in lightning flashes between the two small peaks seems insignificant. As mentioned in this comment, these two small peaks can be regarded as one peak. We have rewritten this sentence. (Line 195)

(3) Low and Middle SHEAR –For simplicity, it might be best to use 850 to 700 hPa shear for low shear and 500 to 400 hPa shear for middle shear.

**Reply**: Thank you for your comment. We have replaced low SHEAR and middle SHEAR with 850 to 700 hPa shear and 500 to 400 hPa in the manuscript and related figure.

(4) L280: "In the afternoon, the relationship between TCLW and T may contain the above two mechanisms, which leads to an insignificant relationship between them"

Is this what you mean?

 "In the afternoon, for TCLW < 0.1 kg m-2, the change in T with increases in TCLW is small as both factors are in play".

**Reply**: Thank you for your comment. Yes, what we want to express is consistent with this comment. We have rewritten this sentence. (Lines: 297-299)

(5) 281: At night, the absence" ◊ "At night, for TCLW < 0.1 kg m-2, the absence"

**Reply**: Thank you for your comment. We have revised this sentence. (Line 297)

(6) L269-296: This paragraph is difficult to follow. Ideally, it should be re-written – perhaps shortened.

Reply: Thank you for your comment. We have rewritten part of this paragraph to make it clearer.

(7) L570: I'm not sure what you mean by "Black lines represent the 1500 m contour line". Are you simply saying that the 1500 m altitude line is shown to highlight the location of the basin?

**Reply**: Thank you for your comment. The black lines represent the 1500 m altitude line. We have revised this sentence. (Lines: 594-595, 601-602 )

**Technical Corrections:**

**Reply**: Thank you for your patience and careful technical corrections to this manuscript. All technical errors have been corrected in this manuscript.

**Referee #3 Report #2**

**Specific Comments:**

**(1)** Lines 12-17: The authors first claimed that "the difference in lightning flashes between the clean and polluted subset was not obvious in the afternoon", however, the authors continued to say that "increasing AOD will lead to an increase in lightning flashes in the afternoon and night". These two statements are contradictory. The abstract is very important for readers, and the authors are expected to write it as clearly as possible.

**Reply**: Thank you for your comment. We have revised this part to make it clearer. (Lines: 12-26)

(2) Lines 232-235: Why are the correlation coefficients between lightning flashes and temperature the same (R = 0.68) in the afternoon and evening? The relationships in Figs. 8a and 8e are not linear relationship, but here it is called a positive relationship? Namely, "Positive relationships can be found in lightning flashes and T both in the afternoon and at night". This description is inaccurate. Please clarify.

**Reply**: Thank you for your comment. Higher surface temperature (T) is conducive to convection and lightning both in the afternoon and at night. Therefore, the correlation between lightning activity and T is positive during the day and night. However, due to the limited number of samples, there may exist errors in the obtained correlation values, resulting in the same correlation values in this manuscript. In fact, based on the results of this manuscript, the relationship between lightning flashes and T in the

afternoon is different from that at night. The non-linear relationship between the lightning flashes and T is more obvious in the afternoon. In future research, we will use more appropriate data to investigate the relationship between them to obtain more accurate results.

We have revised the relevant content in the manuscript to make it more accurate. (Lines 242)

(3) In addition, what is the reason for the above nonlinear relationship?

**Reply**: Thank you for your comment. Higher surface temperature will increase the instability of the atmosphere, which is conducive to the generation of thunderstorms and the increase of lightning flashes. Therefore, when the surface temperature is relatively low, the number of lightning flashes is very small. But exorbitant surface temperature means that a lot of solar radiation reaches the ground without being blocked. It means that there may not be many clouds or even clear days. Therefore, when the surface temperature is too high, there will not be too strong thunderstorms. This may cause a nonlinear relationship between surface temperature and lightning flashes.

(4) Lines 255-258: The authors claimed that "The CG lightning flashes increase with the increase of TCLW when the TCLW is relatively low (<~01 kg m-2), but decrease with the rise of TCLW when its value exceeds

about 0.1 kg m-2". But the explanation given by the authors cannot convince me. Zhao et al. (2020) also discussed the relationship between these two and indicated that it was related to different regions in this region. Please discuss these two different explanations.

**Reply**: Thank you for your comment. Zhao et al. (2020) investigated the relationship between TCLW and lightning density in regions of the Sichuan Basin and the western Sichuan Plateau. Due to the different topography of these two regions, the warm clouds in the plateau area are thinner than those in the basin area. This leads to a smaller value of TCLW in the plateau area than that in the basin area. As found by Zhao et al. (2020), the lightning density is relatively large when the TCLW is equal to about 0.1 kg m$^{-2}$. Our result in this manuscript is similar to theirs. Zhao et al. (2020) pointed out that in the basin region, too much TLCW means robust warm-cloud processes, which are more conducive to the formation of warm rain than ice-phase processes, thereby inhibiting lightning activity. The basin TCLW values of most of their samples are larger than 0.1 kg m$^{-2}$. Therefore, our explanation of the negative relationship between TCLW and lightning flashes when the TCLW exceeds 0.1 kg m$^{-2}$ is consistent with that of Zhao et al. (2020). Because the time resolution and processing method of their data is different from our research, the TCLW values of their samples in the basin area are mostly greater than 0.1 kg m$^{-2}$. Our research reflects the relationship between lightning and TCLW when TCLW is relatively small

(TCLW<0.1 kg m$^{-2}$). We believe that when TCLW is relatively small, it is not easy to trigger warm-rain processes. Under appropriate updraft conditions, TCLW will be transported upward to participate in the ice phase processes in the cloud to form more lightning. This principle should be similar to Zhao's interpretation of the positive correlation between lightning density and TCLW in plateau areas.

(5) Lines 279-284: The results show that "TCLW was negatively correlated with T in the afternoon and at night". And the authors gave two different explanations for this phenomenon. Firstly, "The thicker and wider clouds will block more solar radiation from reaching the ground, thus reducing the surface temperature" (line 278). Secondly, "too much liquid water in the loud may promote a warm-rain process, The precipitation falling to the ground will significantly reduce the surface temperature" (lines 283-284). Please provide the references for these two explanations separately. And which factor is more important?

**Reply**: Thank you for your comment. We made a mistake in this part of the article. In the afternoon, when the cloud liquid water content is relatively high, these two kinds of impacts on temperature exist simultaneously. At night, the former effect disappears due to the absence of solar radiation. We have revised the related content in the manuscript and added several references for these two explanations. We believe that these two impacts

are both important. However, the specific strength of their impact on temperature cannot be quantified in this paper. This is what we need to pay attention to and solve in our future work. (Lines: 291-305)

(6) Line 287: The authors claimed that "This may be because the ice water content in clouds is related to more factors". Please provide these factors and the references accordingly.

**Reply**: Thank you for your comment. We have modified the explanation of the relationship between TCIW and temperature. (Lines: 305-313)

(7) Lines 331: In which region are these analyses conducted? Please clarify.

**Reply**: Thank you for your comment. We have added the specific study region in this sentence.

(8) Lines339-340: What the authors claimed "reduce the solar radiation reaching the ground" is not the "microphysical effects" of aerosols. The usage of proper nouns should be checked in the paper.

Reply: Thank you for your comment. We have revised this mistake in the manuscript.

(9) I grew tired well reading the paper, which is an indication that the writing needs to be improved.

**Reply**: We apologize for the poor language of our manuscript. We worked on the manuscript for a long time and the repeated addition and removal of sentences and sections obviously led to poor readability. We really hope that the flow and language level have been substantially improved.